# Attributes Shape the Embedding Space of Face Recognition Models

**Pierrick Leroy** [* 1] **Antonio Mastropietro** [* 2] **Marco Nurisso** [* 1 3] **Francesco Vaccarino** [1]

## Abstract

Face Recognition (FR) tasks have made significant progress with the advent of Deep Neural Networks, particularly through margin-based triplet losses that embed facial images into high-dimensional feature spaces. During training, these contrastive losses focus exclusively on identity information as labels. However, we observe a multiscale geometric structure emerging in the embedding space, influenced by interpretable facial (e.g., hair color) and image attributes (e.g., contrast). We propose a geometric approach to describe the dependence or invariance of FR models to these attributes and introduce a physics-inspired alignment metric. We evaluate the proposed metric on controlled, simplified models and widely used FR models fine-tuned with synthetic data for targeted attribute augmentation. Our findings reveal that the models exhibit varying degrees of invariance across different attributes, providing insight into their strengths and weaknesses and enabling deeper interpretability. Code available here: https://github.com/mantonios107/attrs-fr-embs.

## 1. Introduction

Deep Learning (DL) models have achieved state-of-the-art performance for Face Recognition (FR) tasks, due to increased availability of curated datasets (Deng et al., 2019b) and improved recognition architectures and losses (Deng et al., 2022). In the challenging open-set scenario, novel face identities can appear at testing time, hence the problem is framed as a metric learning task (Liu et al., 2017). There are three main innovation to reach outstanding FR results in this scenario. The first consists of learning an embedding space representing images such that pictures of the same identity are closer in the embedding domain (Chopra et al., 2005). Second, the usage of a triplet-loss, as in FaceNet (Schroff et al., 2015), where anchors and negative samples are compared to a target image to obtain the gradient driving the embedding map training (Sankaranarayanan et al., 2016). The third consists in equipping the embedding space of an angular dissimilarity, so that it can be assimilated to a hypersphere, e.g., AM-Softmax (Wang et al., 2018a), SphereFace (Liu et al., 2017), CosFace (Wang et al., 2018b), ArcFace (Deng et al., 2019a), AdaFace (Kim et al., 2022).

However, our understanding and interpretation of the structure of the embedding space learned by an FR model remain significantly underdeveloped. Unlike interpretable attributes, deep face representations exist in a high-dimensional space, making it difficult to discern the specific properties they encode (O'Toole et al., 2018). The problem is further aggravated by the potential high risks associated with the social impact of FR technologies. The black-box nature of these embeddings raises concerns about bias, fairness, and security vulnerabilities (Mery, 2022; Gong et al., 2020). Previous studies have attempted to analyze these embeddings from different perspectives: Yin et al. (2019) introduced a spatial activation diversity loss to encourage the network to learn structured face representations, ensuring that different filters respond to semantically meaningful facial parts such as the eyes, nose, and jaw; On the other hand, Lin et al. (2021) proposed the xCos metric, which provides spatially interpretable similarity maps for face verification, whereas (Diniz & Schwartz, 2020) investigates how well the attribute is implicitly learned by neural network layers.

A more recent line of research focuses on reconstructing facial attributes from embeddings to understand what information is retained in the learned representations. Ren et al. (2024) developed a two-stage attribute recovery framework, which reconstructs facial images from the embedding space and then quantifies the significance of facial attributes within the representation. Their analysis indicates that FR models prioritize encoding shape-related attributes (i.e., jawline and facial proportions) over transient factors like expression and head pose. This aligns with findings that deep models tend to discard non-essential variations in favor of stable, identity-specific features (Yin et al., 2019). Other studies focus on the best camera angle to avoid bias in facial im-

---

[*]Equal contribution [1]Department of Mathematical Sciences, Politecnico di Torino, Turin, Italy [2]Department of Computer Science, University of Pisa, Pisa, Italy [3]CENTAI Institute, Turin, Italy. Correspondence to: Antonio Mastropietro .

*Proceedings of the 42$^{nd}$ International Conference on Machine Learning*, Vancouver, Canada. PMLR 267, 2025. Copyright 2025 by the author(s).

age analysis (Choithwani et al., 2023) or to develop an FR system which accounts for the difference between frontal or profile images (Yang et al., 2021).

We can observe that the training procedure and the contrastive nature of the losses aim to achieve invariance of the embedding mapping over transformations of face images belonging to the same identity (Schroff et al., 2015; Chen et al., 2020b; Deng et al., 2022). Past works demonstrated how data augmentation impact feature learning for generic contrastive models (Wen & Li, 2021). In addition, (Chen et al., 2020a) describes data augmentation as transformations of image features that induce invariance in the learned representation. In theory, in the case of the FR task, we want a dependency of the embedding map to the identity of the face image, while keeping invariance to every other image attribute (Schroff et al., 2015). For example, to obtain invariance of FR towards wearing a face-mask, Ge et al. (2024) makes use of a generative-discriminative approach to convert category-aware descriptors into identity-aware vectors. In addition, previous works have made public enriched datasets for training FR models by incorporating variations over a large range of attributes (Moschoglou et al., 2017; Cao et al., 2018).

Taking the metric learning problem of open-set FR to the extreme, an optimal embedding model would map all the images of the same identity to a single point in the embedding space, with images of different identities mapped into distinct points. In practice, this behavior is not observed and might not even be desirable. Indeed, Dan et al. (2024) proposes a method to align the topological structure of the input and the embedding space to improve the generalization of FR models.

We begin our study of the embedding space by noting that there are attributes that orient the relation between identity point clouds. In addition, there is a structure emerging inside the regions corresponding to each embedded identity. Thus, we suggest that the training procedure for FR determines at least two scales at which we can analyze the embedding space. First, the macroscale looks at relations and distances between point clouds of different identities. It is a global scale that looks at all the training images at once, clustered by identity classes. Second, the microscale inspects the inner structure of identity point clouds.

This paper aims to study how the embedding space geometry of FR models is shaped by human-interpretable attributes. First, we examine the macroscale, obtaining insights into the models strategies to solve the FR task. In particular, we provide a procedure to check if attributes shaping the relations inter-identities are the ones most deterministically linked to an identity. Our results over the comparison between FaceNet and ArcFace suggest that the embedding of the former has a higher structural dependency of the

attributes linked to identities. Then, we focus on the microscale, and we propose an invariance energy measure to quantify the invariance of the embedding model to each attribute. Since the embedding model at the microscale should filter out transformations of face images that preserve the identity, we hypothesize that the attribute invariance at the microscale level indicates the quality of the FR performance. We validate our hypothesis by fine-tuning a number of state-of-the-art FR models on synthetic, generated data, varying a set of interpretable attributes one by one in a controllable manner through the GAN-Control model (Shoshan et al., 2021). The results show that the fine-tuning on a specific attribute increase the invariance metric on the same attribute.

## 2. Multi-Scale Geometry of the Latent Space

We establish a formal ground for the FR task as assumed by Deep FR models and losses. We assume an open set protocol, that is, the testing identities do not necessarily cover the training identities; hence, we need to map faces into a discriminative embedding space, so that the FR task gains similarities with the metric learning problem. For other FR protocols, we point the reader to Liu et al. (2017). Furthermore, we assume a trained FR model on a sufficiently representative dataset, e.g., MS1MV3 (Deng et al., 2019b), and we inspect the embedding space to describe its geometric structure induced by the architecture and the loss.

Let $X = \mathbb{R}^k$ be the image space and $\mathcal{D} \subset X$ be the space of face images, partitioned by the set of labeled identities $I$, $\mathcal{D} = \cup_{i \in I} \mathcal{D}_i$, where $\mathcal{D}_i = \{x^{(i)}\}$ is the subset of face images associated with identity $i$. The dataset of face images $D$ represents a sample from $\mathcal{D}$, and similarly, $D_i \subset \mathcal{D}_i$. Let $f : X \to E$ be a pretrained face embedding model, mapping images $X$ to a space of vector embeddings $E$, immersed in an outer space of dimension $p$, i.e. $E = \mathbb{R}^p$, equipped with a distance (or dissimilarity) function $d : E \times E \to \mathbb{R}$. For example, the embedding space of FaceNet (Schroff et al., 2015) is equipped with the Euclidean distance, whereas ArcFace (Deng et al., 2022) with cosine dissimilarity. We define the set of embeddings $S_i = f(\mathcal{D}_i)$ as an identity space. Likewise, we call $P_i = f(D_i)$ as an identity point cloud, obtained by sampling from $S_i$ or by sampling from $D_i$ and embedding through $f$ in $E$.

Furthermore, at inference time, we assume that there is a classification rule $r : E \to I \cup \{u\}$, such that if an embedding corresponding to an image $e = f(x)$ is sufficiently close to a $P_i$, then $r(e) = i$; we anticipate that a $x$ could be assigned to no identities, in which case the rule associates the unidentified identity $u$ to the embedding $e$. Since the classification rule depends on the model and loss employed, we leave the rule as an abstract function. However, we imply that $r$ makes use of the distance (or dissimilarity) of $E$. In this setting, a $i$-decision region $R_i = r^{-1}(i)$ is the subset

*Table 1.* Intra-class and inter-class distance on the LFW dataset

| Model | Architecture | $\bar{d}$ | $d_b$ |
|---|---|---|---|
| ArcFace | ResNet18 | 0.327 | 0.994 |
| ArcFace | ResNet50 | 0.290 | 0.996 |
| AdaFace | ResNet18 | 0.370 | 0.995 |
| SphereFaceR | iResNet100 | 0.276 | 0.993 |
| FaceNet | iResNetv1 | 0.670 | 1.376 |

of $E$ such that the projected embeddings by $f$ are classified by $r$ as identity $i$. Note that $R_i$ could be different from the identity space $S_i$, and, in general, the decision regions of the training identities do not partition $E$.

Loosely speaking, a Deep FR model is trained by optimizing a loss function pushing the distance inter decision regions higher than the size of each region. The (inter-class) distance between decision regions can be estimated through the corresponding distance between identity point clouds: $d_b(P_i, P_j) = \frac{1}{|P_i||P_j|} \sum_{e^{(i)} \in P_i, e^{(j)} \in P_j} d(e^{(i)}, e^{(j)})$. Similarly for the (intra-class) distance within decision regions: $\bar{d}(P_i) = \frac{1}{|P_i|^2} \sum_{e^{(i)}, e^{(j)} \in P_i} d(e^{(i)}, e^{(j)})$ (Liu et al., 2022; Deng et al., 2022). Thus, after training, we can assume that the distance within is much smaller than the distance between point clouds, i.e. $\bar{d}(i) \ll d_b(i, j)$ and $\bar{d}(j) \ll d_b(i, j)$ for $i, j \in I$. Indeed, Table 1 shows that there is at least a factor of 2 between $\bar{d}$ and $d_b$ for FR models with various architectures, losses and distance metrics on the LFW dataset (Huang et al., 2008).

From the assumptions made, if we look at the mapping of the dataset $\mathcal{D}$ to the embedding space $E$, we ought to see separate point clouds, each corresponding to a single identity $i$. This geometry is expected, as it is a direct effect of the contrastive loss used in training: embeddings belonging to the same identity are pulled together, while simultaneously being pushed away by the embeddings of other identities. Thus, the embedding space $E$ of $f$, given the dataset $D$, has two natural levels of geometry corresponding to two scales of observation: (i) the *microscale* delves into the structure of each individual identity point cloud, whereas (ii) the *macroscale* pertains to the way various identity point clouds, now seen as single points, are arranged in $E$.

It is important to recognize that, in terms of pure performance, the microscale is not crucial as long as all face images labeled $i$ end within the correct region $R_i$, and $R_i$ does not contain elements of $D_j$, $j \neq i$. In particular, a model should possess an invariance property with respect to the attributes of facial images. Hence, if we transform a face image $x$ over a specific attribute through a transformation $t$, while preserving the identity $i$, e.g., as of $t(x) = x' \in D_i$, then a performing model should satisfy an invariance property $f(t(x)) \approx f(x)$. Sec. 2.1 formalizes this property. In

fact, in the extreme case that a model $f$ is such that for each $i \in I$, the identity cloud trivially collapses $f(D_i) = \{e^{(i)}\}$ and such that for distinct $i, j \in I$, $e^{(i)} \neq e^{(j)}$, then it is easy to define a rule $r$ to achieve the best recognition.

However, in real settings, the many losses proposed to tackle the FR task paired with the various architectures show non-trivial geometry, without ending in strong overfitting or mispredictions. This results in each identity point cloud possessing a structure that preserves some properties of the image space. Moreover, as far as the global scale is concerned, the triplet loss does not explicitly encourage any particular disposition of the identities in the embedding space, meaning that any random disposition of the point clouds, as long as they are sufficiently far away from each other, is equally valid.

Interestingly, we find that real face recognition models exhibit strong global organization, such as the separation of male pictures from female pictures as shown for example in Figure 1. As it is not explicitly aimed at or due to the loss, this structure must be an emergent phenomenon due to the synergy between initialization, optimizer, architecture, and data. This fact is suggested also by Hill et al. (2019): the authors found that a hierarchical structure emerges by analyzing the embedding space of a CNNs model for the FR task trained using a triplet loss (Sankaranarayanan et al., 2016) and looking at interpretable attributes of face images.

### 2.1. Attribute Invariance

We call image attribute $a\colon \mathcal{D} \to M_a$ a function recognizing an interpretable property on face images; we name $M_a$ the modality space for $a$. We denote by $A$ the set of attributes we want to recognize on each image. We can distinguish many attribute classes: for example, Hill et al. (2019) cites viewpoint, illumination, facial expression, or appearance. Some attribute $a$ may be strongly related to facial identity, such that $a(x)$ is constant on each $\mathcal{D}_i$; on the opposite, others can be derived from the image properties, like contrast or brightness, or can vary between pictures of the same identity (e.g, face mask, eyeglasses). Depending on the attribute, $M_a$ can be a manifold or just a set. For instance, we can describe *blonde hair* as a binary attribute that has value 1 if and only if the picture shows a face with blonde hair $a\colon x \in \mathcal{D} \to M_{\text{blonde}} = \{0, 1\}$. *Age* is a continuous attribute that associates to a picture the age of the person depicted at that moment. In this case, we can assume $M_{\text{age}} = (0, +\infty)$. Finally, *hair color* is a continuous attribute that associates to a picture the person's hair color, belonging to a suitable color manifold $M_{\text{color}}$.

In most cases, the attribute function may not have an explicit analytic definition, and we may have access to its values only through a dataset. For example, a widely used FR dataset as CelebA provides us with binary labels for a list

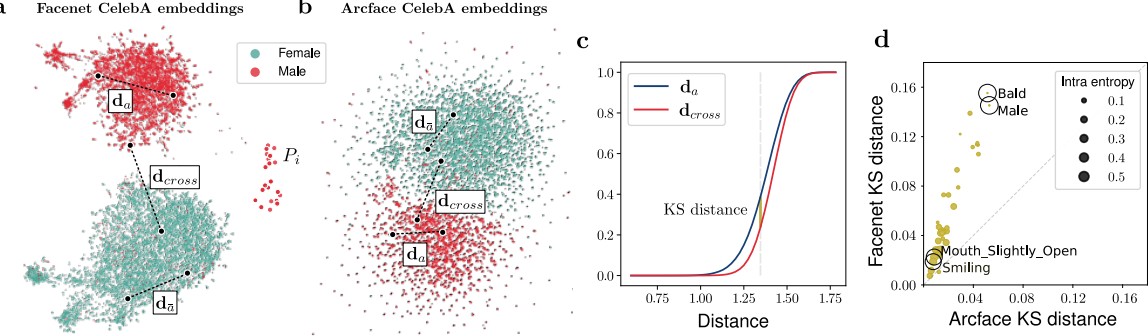

*Figure 1.* UMAP projection (McInnes et al., 2018) of the Facenet (**a**) and Arcface (**b**) embeddings of CelebA. The colors show a clear separation between images depicting males and females. Note that the UMAP projection does not accurately capture the distances in the original $E$ space. **c.** For each binary attribute, we compute the distributions of distances inside and across the two modalities, and compute their Kolmogorov-Smirnov distance. **d.** Model comparisons: each point corresponds to an attribute whose coordinates encode the KS statistic for ArcFace and for FaceNet. The size of each point is proportional to the average intra-entropy of the corresponding attribute.

of 40 attributes (Liu et al., 2015), even if for some of them the definition is fuzzy (e.g. blonde hair). Moreover, we might not have direct access to the modality space $M_a$ but, having fixed an image $x$, we could have a function to transform $x$ while varying the value of $a(x)$ in a controllable way. Inspired by the theory about data augmentation (Chen et al., 2020a), for a set of attributes $a \in A$ we can define groups of transformations $T_a = \{t_a \colon \mathcal{D} \to \mathcal{D}\}$, equipped with the composition operator $\circ$. For example, for the binary attribute $a$ having modalities $M_{\text{blonde}} = \{0, 1\}$, we can think of $T_a = \{z, w\}$: $z$ is the zero element, $z(x) = x$, whereas $w$ is a transformation from "blonde" to "not blonde" and vice versa, i.e. for each $x \in \mathcal{D}$, $a(w(x)) = 1 - a(x)$, with the constraint $(w \circ w)(x) = z(x) = x$. Furthermore, we define the action $\alpha_a$ of a group of transformations $T_a$ on $M_a$:

$$\alpha_a \colon T_a \times \mathcal{D} \to \mathcal{D} \qquad (1)$$
$$(t_a, x) \to t_a(x)$$

We recall that by the property of the action of a group on a set: (1) for the group zero element $z \in T_a$, $\alpha_a(z, x) = z(x) = x$, $\forall x \in \mathcal{D}$; (2) the action is compatible with the composition $\circ$ of transformations, i.e. for every $t_a, s_a \in T_a$, $\alpha_a(t_a \circ s_a, x) = \alpha_a(t_a, \alpha_a(s_a, x))$.

We can weaken the invariance property by asking that the embedding of the transformed images stay close to the base image $f(t_a(x)) \approx f(x)$. As mentioned in Sec. 2, the microscale of an identity point cloud is measured by the within distance $\bar{d}(P_i)$. In particular, we state that the attribute $a$ satisfy an approximate invariance property if, for all $t_a \in T_a$, $d\left(f(t_a(x)), f(x)\right) < \delta\, \bar{d}(P_i)$, where $\delta \in [0, 1]$ is an hyperparameter which determines the allowed displacement of $t_a(x)$. Hence, it is meaningful to measure the approximate invariance of an embedding model $f$ relative to a set of attributes $A$ and their respective group of transformations.

In general, note that a transformation of $a$ could impact multiple attributes at the same time (e.g. age and hair color). Thus, by measuring the invariance over a set of attributes we can observe their entanglement in $E$.

## 3. Macroscale Analysis

In this section, we analyse the structure of the embedding space at the macroscale, looking at the possible interaction of the identity point clouds $P_i$ at inter-identity distance $d_b$. Given a model $f$, a dataset $D$, and an attribute $a$, it is interesting to determine whether $a$ influences $E$ across different $P_i$. We propose a simple methodology to partially answer this question when $a$ has discrete values. Using this approach, we analyze a high quality subset of CelebA consisting of $|I| = 1965$ identities, with $|D_i| \approx 30$ and 40 binary attributes (e.g. "eyeglass", "smiling", "male") for five FR models: FaceNet, AdaFace, SphereFaceR and two versions of ArcFace.

### 3.1. Attribute Dependence in Embedding Space

We present a simple procedure to test if an attribute has a structural impact on $E$ relying only on distances/dissimilarities. More precisely, we propose to test whether conditioning a $P_i$ on the same attribute modality $m_a$ of an attribute $a$ changes the distance distribution compared to conditioning on different modalities. In other words, we test the hypothesis:

$$\mathcal{H}_0 \colon F_a(d(e_1, e_2)) = F_{\bar{a}}(d(e_1, e_2))$$

where $e_i = f(x_i)$ are embeddings, $F_a$ denotes the cumulative distribution function (CDF) in the case when the images have the same modality for attribute $a$, i.e., $a(x_1) = a(x_2) = m$, for some $m \in M_a$, and $F_{\bar{a}}$ is the CDF when the modalities are different. We use the Kolmogorov-

*Table 2.* Summary of Spearman correlation results.

| Model | Architecture | Correlation | P-value |
|---|---|---|---|
| FaceNet | iResNetv1 | -0.626 | $\leq 10^{-4}$ |
| ArcFace | ResNet50 | -0.566 | $\leq 10^{-3}$ |
| ArcFace | ResNet18 | -0.619 | $\leq 10^{-4}$ |
| AdaFace | ResNet18 | -0.656 | $\leq 10^{-4}$ |
| SphereFaceR | iResNet100 | -0.606 | $\leq 10^{-4}$ |

Smirnov test from Massey Jr (1951) (KS-test) to check the equality of distributions as stated by $\mathcal{H}_0$. We expect the attributes of CelebA to influence the embedding space of our FR models, precisely because these attributes are facial features, useful to solve the FR task.

To test $\mathcal{H}_0$ for an attribute $a$, we first sample *modality point clouds* $Q_m = \{e \in E \mid e = f(x), a(x) = m\}$. Note that the $Q_m$ span embeddings across $P_i$, with possibly multiple points per identity. Since we are only interested at the macroscale, we subsample each $Q_m$ to keep at most a single representative per identity. To compare between modalities of the same attribute, we further subsample such that, fixed $a$, all the $Q_m$, $m \in M_a$ have the same cardinality $q$. Then we measure: the intra-distances $d_m = \{d(e_1, e_2) \mid e_1, e_2 \in Q_m, e_1 \neq e_2\}$, representing the pairwise distances between points in a $Q_m$, and the inter-distances $d_{\overline{m}} = \{d(e_1, e_2) \mid e_1 \in Q_m, e_2 \notin Q_m\}$, representing distances between points of different $Q_m$. Finally, for each modality, we perform a KS-test on the empirical distributions of $d_m$ and $d_{\overline{m}}$, the statistic of the test being the KS-distance depicted on Figure 1c. For CelebA, $m \in \{-1, 1\}$ and we reject $\mathcal{H}_0$ for a large majority of attributes at the confidence level $0.001$ (see detailed p-values on Figure 10 in Appendix C). Furthermore, to summarize the KS-statistics at the attribute level, we compute the mean over its modalities: $KS_a = \text{mean}_{m \in M_a}(KS_m)$.

### 3.2. Geometry and Discriminative Power

**Intra-entropy is a proxy for discriminative power.** In FR tasks, the discriminative power of an attribute reflects its contribution to prediction performance and information gain. For instance, *smiling* has likely low discriminative power since knowing whether someone is smiling provides little help in identifying them, whereas *eye color* is more distinctive and thus has higher discriminative power. We propose a way to detect if the discriminative power of attributes is reflected by the geometry in embedding space. Given an attribute $a$, we approximate its discriminative power by its *intra-entropy* $H_a$: first for each identity $i$ we compute $H_a^i = H(\{a(x) \mid x \in \mathcal{D}_i\})$, where $H$ is the empirical entropy. Then, the intra-entropy is simply an average over identities $H_a = \sum_{i \in D} H_a^i$. In CelebA, we observe that attributes such as *male* or *bald* have low intra-entropy,

whereas *smiling* or *mouth slightly open* have high intra-entropy. This finding aligns with intuition, since the last two can easily vary for a fixed identity, while it is less likely for the first two. See Appendix C for the intra-entropy of each attributes and other summaries.

**Linking intra-entropy with KS statistics.** For ArcFace and FaceNet on CelebA, we find significative negative Spearman correlations between the intra-entropies $H_a$ and the KS statistics $KS_a$ (Table 2). This confirms that the attributes shaping $E$ the most are the ones most deterministically linked to an identity.

**Additional observations.** In light of this analysis, we can make additional observations. On Figure 1d, we propose a comparative analysis by plotting two models against each other. The point cloud being above diagonal informs us that FaceNet has higher KS statistics, hinting at a bigger structural dependency on the attributes. While there are slight variations, the two models tend to converge on which attribute have the bigger structural impact, as attribute are ranked similarly, as in (Li et al., 2015). All KS statistics are significantly lower than 1, meaning that no attribute is associated with distinct clusters of points, as it is the case for identities. Note also the correlation from Table 2 of intra-entropy and KS statistics visualized as larger dots tend to concentrate towards the origin. Lastly, modalities are not necessarily symmetric and can have unbalanced KS statistics (example: *bald*, Figure 10 in Appendix).

## 4. Microscale Analysis

After analyzing how identity attributes shape the macroscale geometry, we aim to see if and how much the microscale is influenced by the the same attributes, or if the embedding model is invariant to them.

When working with categorical attributes, given that the distance in the embedding space $E$ is used for the decision rule, it is enough to check the displacement induced by the attribute variation. Categorical attributes are then readily comparable because we can assess their associated changes in the embedding space when we change their modality in the attribute space. With continuous attributes, however, it is harder because there is no natural way to compare the amount of change for an attribute $a_1$ with another attribute $a_2$ due to their possibly incomparable scales. For example, if one fixes a change in *illumination*, it is impossible to choose an "equivalent" change in *hairstyle*. To approach this issue, we propose an *energy* measure which is independent of the attribute scale. The general idea is to study the vector field in $E$ induced by the infinitesimal variations of an attribute $a$, as we see in Figure 2. Every vector in the vector field is normalized to unit norm so that we can ignore the effect of

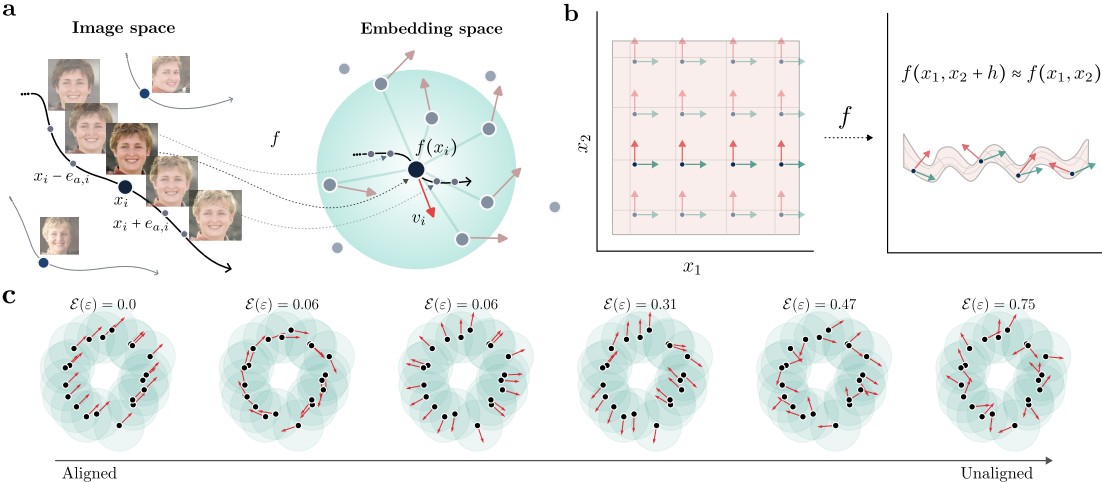

*Figure 2.* **a.** Intuitive depiction of the computation of the invariance energy. The continuous variation of the *hair color* attribute results in a curve passing through each embedding. Our energy measure computes the dissimilarity between the normalized tangent vectors to these curves at points which are closer than a threshold $\varepsilon$. **b.** Visualization on how the model achieves approximate invariance w.r.t. dimension $x_2$ by irregularly folding the space. **c.** Examples of unit-norm vector fields on a simple 2-dimensional point cloud ordered by their $\mathcal{E}$.

scale, leaving only direction and orientation.

In detail, let us consider a one-parameter group $G$ (Varadarajan, 2013) representing the action of the group of real numbers $t \in \mathbb{R}$ varying the attribute $a$, $\alpha_{a,t} : X_a \to X_a$ , $\forall t \in \mathbb{R}$ with $\alpha_{a,0}(x) = x$. This situation corresponds, for example, to the one provided by GAN-Control, if the attributes are changed through monodimensional sliders. Fixed an image $x \in D_i$ we define the vector field at the embedding $e = f(x) \in P_i$ as $v_a(e)$ with:

$$v_a(e) := \frac{\tilde{v}_a(e)}{\|\tilde{v}_a(e)\|}, \quad \tilde{v}_a(e) := \frac{d}{dt}\bigg|_{t=0} f \circ \alpha_{a,t}(x) \in T_e E,$$
(2)

where $T_e E$ is the tangent space of $E$ at $e$, which we identify with $\mathbb{R}^p$. If $\|\tilde{v}_a(e)\| = 0$, we write $v_a(e) = 0$.

### 4.1. Energy Measure

Our conjecture is that, if the FR model has learned to be approximately invariant to an attribute $a$, it will do so by trying to collapse the directions associated to $a$ in the input point cloud $D_i$ by folding and distorting the space in a complex way. This folding, akin to crumpling a sheet of paper, will push together points which were further away in the input space. If this is the case, we can observe the behavior by computing the "roughness" of the attribute vector field $v_a$, quantifying the lack of alignment of each vector $v_a(e)$ with the vectors at close-by points. An intuitive visualization of this fact is depicted in Figure 2b, where we see how the dimension associated to attribute $x_2$ is contracted non-linearly, pushing closer vectors which were far away. On the other hand, the FR model will be sensitive to the attribute if $v_a(e)$ is aligned with the vectors associated with close-by points.

In this case, the attribute's continuous variation will result in a *locally constant* displacement of the embeddings, that is, close embeddings are moved in the same direction.

In detail, fixing a scale parameter $\varepsilon \geq 0$, we define the *invariance energy* metric associated to the identity space $S_i$ and vector field $v_a$ of $a$, as the expected cosine of the angle between vectors at points $e, e' \in E$ which have distance less or equal than $\varepsilon$.

$$\mathcal{E}_{a,i}(\varepsilon) := \mathbb{E}_{e \sim S_i} \mathbb{E}_{e' \sim B_\varepsilon(e) \cap S_i} \left[ 1 - v_a(e)^\top v_a(e') \right], \quad (3)$$

where the first expectation is taken over a uniform sampling of $S_i$, and the second over a uniform sampling of the closed ball with center $e$ and radius $\varepsilon$ in the distance/dissimilarity function $d$, intersected with the identity space $B_\varepsilon(e) \cap S_i$. Since we have access to $S_i$ through the identity point cloud $P_i$, we compute an estimate of the invariance energy over the embedding of $P_i$. The name "energy" comes from the fact that, when we approximate Equation (3) on a given $P_i$, the energy $\mathcal{E}(\varepsilon)$ is closely reminiscent to the Hamiltonian of the $n$-vector model (Stanley, 1968) describing the evolution of a system of pairwise-interacting $n$-dimensional spins.

As we see in Figure 2c, if the vector field is perfectly aligned $v_a(e) = v_a(e') \ \forall e, e' \in P_i$, we will have minimum energy $\mathcal{E}_{a,i}(\epsilon) = 0 \ \forall \varepsilon > 0$. It will have low values when $v_a$ is locally aligned, while it attains the highest values for disordered random-looking vector fields. The maximum energy value is $\mathcal{E}_{a,i} = 2$, with $\mathcal{E}_{a,i} = 1$ meaning that the vectors are on average orthogonal to one another.

**Toy model.** To validate the meaningfulness of our energy definition (3), we study a simple toy model where approximate invariance is achieved by a multi-layer perceptron

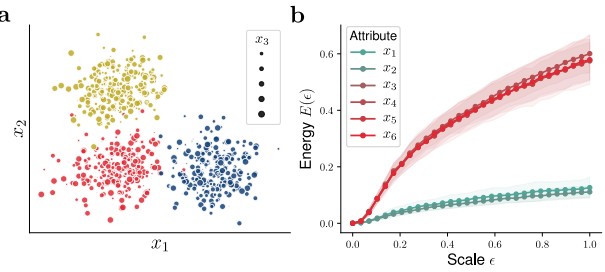

*Figure 3.* **a.** The generated dataset for the toy model. **b.** The energies associated to each input dimension for different scales, over 30 neural network trainings.

(MLP). We generate 3 6-dimensional blobs of 250 points each, such that their first two features $x_1, x_2$ are distributed according to three separated Gaussians as in Figure 3a. We name these dimensions *useful* because they can be employed by the MLP for classification. The values of the last 4 *useless* features $x_3, \ldots, x_6$, instead, are independently sampled from uniform distributions between -5 and 5.

We train to convergence a simple three-layered MLP to classify the blob each point belongs to. We see how the information about the blob separation is present only in the first two dimensions, meaning that the model will need to learn to be invariant to the last four. We pick a range of scales $\varepsilon \in [0, 1]$ and compute the energy measure on the embeddings provided by the 16-dimensional penultimate layer. The results, shown in Figure 3b, clearly show how the energies are nicely divided into two groups, with the useless dimensions (attributes) having a clearly larger energy than the useful ones at all scales.

### 4.2. Energy of Pretrained FR Models

To compute the invariance energy for actual FR models, we need a way to sample face images with locally modified attributes, i.e. to mimic the action of group transformations over attributes. For this, we turn to generative modeling. GAN-Control (Shoshan et al., 2021) uses attribute classifiers during training to create a disentangled latent space. Then, it learns to map attribute vectors to their corresponding latent subspace. This allows users to generate images in a controllable manner, specified with interpretable attributes. We also experimented with ConfigNet (Kowalski et al., 2020) and Arc2Face (Papantoniou et al., 2024) but GAN-Control provided the best balance for our needs in terms of controllability, generation quality and efficiency. The disclosed pretrained GAN-Control model lets us tinker with the following 5 attributes, fixing an identity: pose, age, hair color, illumination and expression. We use GAN-Control to generate vast point clouds of identities of a total number of over 120K embedded images sampled along curves where only one attribute changes, as portrayed on

Figure 2a. It is important to make sure the generated point cloud is perceived by FR models as a single identity because we want to conduct a microscale analysis. We do that by verifying that distances between embedded images and their single attribute variations correspond to the order of magnitude of the intra distances reported on Table 1. Concretely, that assures us that the curves we are interested in do not represent traversals spanning more than a typical identity. While GAN-Control allows us to generate images varying high-level attributes, we also include 3 low-level data augmentations: brightness, hue and image quality, where the latter amounts to different levels of blurring and contrast enhancement.

We now take three different pretrained FR architectures, namely FaceNet, ArcFace, AdaFace, and compute the invariant energies of the 8 controllable attributes described above for GAN-Control. We do this averaging over 40 synthetic identities $D_i$, for multiple $\varepsilon$-scales, to obtain a multiscale fingerprint of each model as a whole. Given that the embedding spaces of the three models are equipped with different metrics (Euclidean and cosine dissimilarity), each scale $\varepsilon$ is chosen to be relative to the average distance $\bar{d}$ between pairs of points in each $P_i$.

In Figure 4a, we show the results of this analysis. First, we observe that the set of attributes we chose attains energy values across a wide range, signifying the presence of different behaviors. In terms of the relative positioning of the attributes, we find that all models are most invariant to lower-level attributes like "contrast" and "illumination", while they achieve the lowest values at complex attributes like "head angle" and "age", signifying more sensitivity. We note that this results align with the intuition of complex attributes being less "filtered out". Learning to be invariant to age, in particular, means that the model cannot rely on more lower-level identity features, like the hair color or the exact face proportions.

From Figure 4b, moreover, it is clear how FaceNet achieves a significantly lower energy across all attributes, at all scales. This, in line with the macroscale analysis of Section 3, suggests that the geometry of FaceNet's embedding space is more sensitive to the attributes also at the microscale of single identities. ArcFace and AdaFace, being similar models, have similar values, with AdaFace having slightly lower energy (more sensitivity) on the attributes related to color, i.e., "Hair color" and "Hue".

### 4.3. Invariance Energy on Fine-Tuned Models

To assess the capability of our energy measure to capture the invariance of FR models, we ask if the invariance observed through the energy measure agrees with the effect of data augmentation on single attributes. In fact, by training a model with augmented data coming from the action $\alpha_a$ of at-

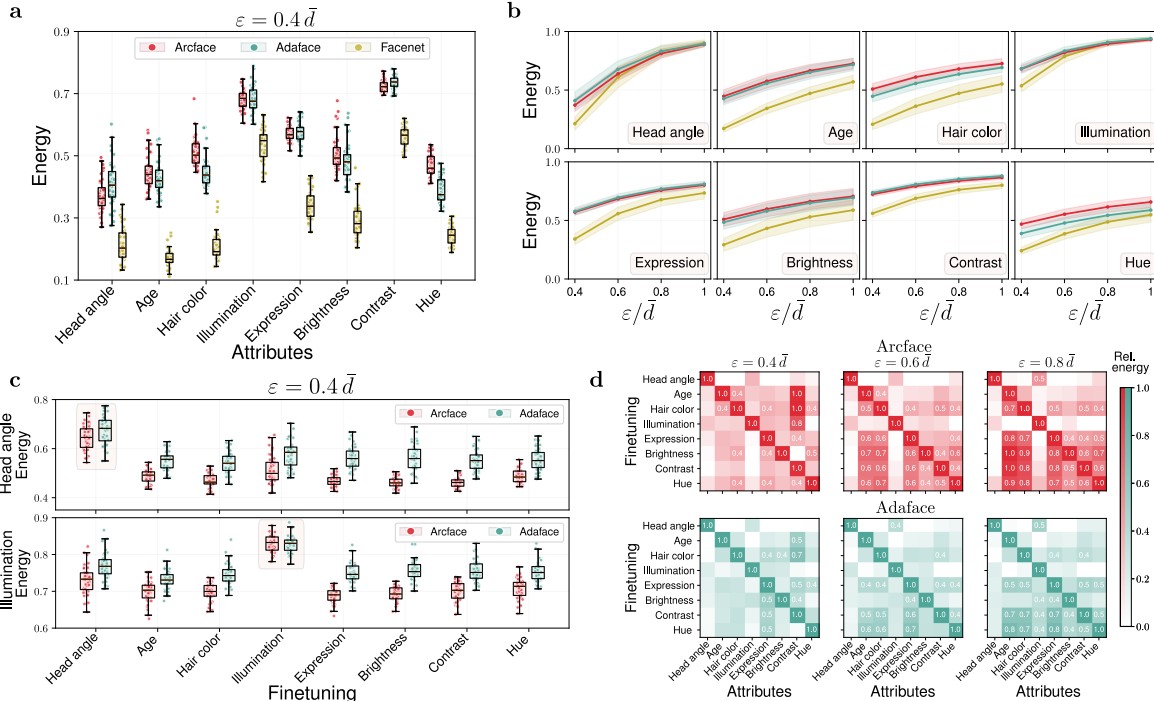

Figure 4. **a.** Energy distributions for each attribute and each FR model, computed across 40 synthetic identities at the scale $0.4\,\bar{d}$. **b.** Average energy for each attribute and FR model when we vary the scale from $0.4\,\bar{d}$ to $\bar{d}$. **c.** The distribution of the *Head angle* (top) and *Illumination* energies after all the fine-tunings (x-axis) of ArcFace and AdaFace. The highest energies, which are attained ad the fine-tuned attribute, are highlighted. **d.** The post fine-tuning energy of every attribute and every fine-tuning under min-max rescaling over the fine-tunings (rows).

tribute $a$, we can expect its internal representation to become more invariant to $a$ (Chen et al., 2020a;b). Our goal is to fine-tune FR models with augmented data for each attribute, and observe the resulting change in the energy measure. In detail, the baseline ArcFace and AdaFace models are fine-tuned on monovariant datasets of 10K identities created with GAN-Control and with a single attribute variation (e.g. only variations in illumination). We also generate validation and test sets containing all possible variations. However, since ArcFace and AdaFace have parametric losses specific to train labels, they can't be directly computed on a hold-out sets in the open-set protocol. We slightly adapt these losses to make them parameter-free (details in Appendix F.1) and so we are able to measure them on validation and test sets.

We fine-tune multiple times our baselines on monovariant datasets and use the parameter-free loss to determine the best checkpoint. During fine-tuning, we track key metrics on the validation and test sets and on CelebA (cf Appendix F). After the process, we obtain a family of specialized models, one for each attribute. Let $f_a$ be the FR model fine-tuned on attribute $a$ and $\mathcal{L}_a$ denote the parameter-free loss measured on a hold-out monovariant set $D_a$ containing only variations of attribute $a$. Then, we observe that $\mathcal{L}_a(f_a) < \mathcal{L}_a(f_{a'})$, $\forall a \neq a'$, i.e. each model $f_a$ has effec-

tively learned to be more invariant to the variations of $a$ than the other models $f_{a'}$. We observe that the specialization induced by fine-tuning is not completely disentangled, as fine-tuning on one attribute (e.g., age or brightness) may increase performance on another, like hair color, more than on orientation. Fine-tuning metrics such as learning curves are presented in Appendix F.

We expect the invariance energy to capture the shift in internal representation caused by the fine-tuning. Figure 4c, shows two examples ("head angle" and "illumination") where, fixing an attribute $a$, the fine-tuning on $a$ is the one increasing the energy value the most. This holds for both ArcFace and AdaFace in the same way. Figure 4d summarizes the results by showing the per-attribute energies min-max rescaled across the fine-tunings. Notice how, for both ArcFace and AdaFace, the maximum value of each column is achieved at the diagonal, meaning that $\mathcal{E}_a$ is increased the most when fine-tuning on the same $a$.

## 5. Conclusion

In this paper, we explored the geometric properties of the embedding space of FR models, emphasizing how human-interpretable attributes influence both macroscale and mi-

croscale structures. For macroscale, we focused on categorical attributes, and proposed a simple approach to assess their influence and discriminative power using distribution distance. For the microscale, we focused on continuous attributes and introduced an energy measure to quantify the degree of invariance of a model to an attribute. Our proposal effectively quantifies the sensitivity of FR models to different attributes, highlighting variations across models, and demonstrating how they encode attributes with varying degrees of invariance. The measurements agree with fine-tuned models, confirming that targeted augmentation enhances attribute-specific invariance. Our findings provide deeper interpretability of FR embeddings and suggest new avenues for improving the robustness of FR systems. Although we focus on FR, our ideas could be applied in other metric learning contexts, provided attributes are available.

## Acknowledgements

M.N. acknowledges the project PNRR-NGEU, which has received funding from the MUR – DM 352/2022.
This study was carried out within the FAIR - Future Artificial Intelligence Research and received funding from the European Union Next-GenerationEU (PIANO NAZIONALE DI RIPRESA E RESILIENZA (PNRR) – MISSIONE 4 COMPONENTE 2, INVESTIMENTO 1.3 – D.D. 1555 11/10/2022, PE00000013). Work supported by the European Union's Horizon Europe research and innovation program for the project FINDHR (g.a. 101070212). This manuscript reflects only the authors' views and opinions; neither the European Union nor the European Commission can be considered responsible for them.

## Impact Statement

The risks associated with FR models extend beyond technical inaccuracies. Biases in the embedding space can lead to disparities in recognition accuracy across demographic groups, raising ethical concerns in applications such as surveillance, border control, and authentication systems. The reliance on black-box embeddings also creates vulnerabilities, where adversarial manipulations or synthetic alterations can exploit weaknesses in the learned representation. These issues highlight the need for interpretability-driven approaches that can assess and guide the structure of FR embeddings. A clearer understanding of the embedding space of FR models is essential for ensuring their reliability, fairness, and security. Given that these models learn high-dimensional feature representations, their decision-making process remains largely opaque, making it difficult to assess whether embeddings capture identity-related features robustly or if they encode unwanted biases. Investigating the structure and invariance properties of embeddings can help uncover hidden dependencies on attributes such as

age, pose, or illumination, which may influence recognition performance in unintended ways. Greater transparency in embedding learning can contribute to fairer and more accountable biometric technologies, ensuring that face recognition systems function reliably across diverse populations while mitigating potential misuse.

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

# A. Datasets

**LFW(Huang et al., 2008)**    LFW is a dataset of approximately 13K images with no additional attributes beyond identity in its standard version. The number of images associated with each identity varies between a handful and several hundred. We use this dataset as a sanity check dataset for the models. Indeed, we expect the performance of all models to be almost perfect on the face verification (Huang & Learned-Miller, 2014) task (classifying matching and non-matching pairs), which is saturated on this dataset. We show examples of distance distributions in Figure 5 and metrics in Table 3

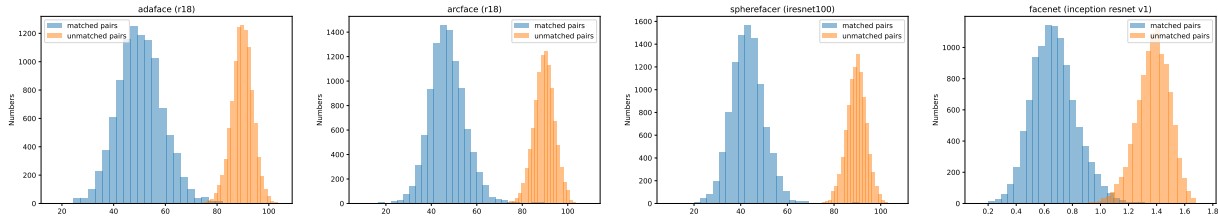

*Figure 5.* Distribution of distances on the face verification task on random pairs extracted from LFW for AdaFace, ArcFace, SphereFaceR and FaceNet. The models where the embedding space is equipped with cosine distance (AdaFace, ArcFace, SphereFaceR) show the distance in degrees.

| model | best accuracy | mean intra distance | mean inter distance |
|---|---|---|---|
| ArcFace (r18) | 0.9974 | 0.3270 | 0.9944 |
| ArcFace (r50) | 0.9991 | 0.2896 | 0.9960 |
| AdaFace (r18) | 0.9953 | 0.3697 | 0.9950 |
| SphereFaceR (iresnet100) | 0.9983 | 0.2760 | 0.9933 |
| FaceNet (iresnet v1) | 0.9851 | 0.6702 | 1.3763 |

*Table 3.* Models metrics. The best accuracy corresponds to the estimation of accuracy at the optimal threshold.

**CelebA(Liu et al., 2015)**    We use the CelebA dataset to conduct an analysis of the attributes at the global scale, that is, across identities. Since the finetuning is done on generated data, we also track a few metrics on CelebA during our finetuning process to keep track of the dynamics on real data. This dataset contains over 200K images and 10K identities, as well as 40 binary attributes (e.g., "eyeglass", "smiling", "male"). To extract faces, we run the Retina Face(Deng et al., 2020) face detector. We encountered common data quality and processing issues:

1. Multiple faces detected on an image.
2. No face detected on an image.
3. False negatives: for a given identity, there are sometimes mislabels i.e. images of different persons.

We took some steps to manage these problems and create a higher-quality subset since we don't need all the data. We address issues 1) and 2) above by discarding images with not exactly 1 face detected. We also restrict ourselves to the identities with more than 30 images. Then, we apply statistical filtering to remove obvious outliers (cf. Section 3.2 in (Deng et al., 2019a)): first, we project images into the embedding space using face recognition models, and then we remove those points that are far from all other points. We verify the improvement of statistical filtering by comparing simple clustering with the identity point clouds. We do this statistical filtering using ArcFace and FaceNet.

After these steps, we reduce CelebA to a subset containing around 55K images of ∼30 images by identity. All analyses are made on the higher-quality subsets.

## B. Models

| Model | Architecture | Metric | Train set | Images (M) | Source repository |
|---|---|---|---|---|---|
| FaceNet | iResNetv1 | euclidean | VGGFace2 | 3.31 | `davidsandberg/facenet` |
| ArcFace | ResNet50 | cosine | MS1MV3 | 5.18 | `deepinsight/insightface` |
| ArcFace | ResNet18 | cosine | MS1MV3 | 5.18 | `deepinsight/insightface` |
| AdaFace | ResNet18 | cosine | VGGFace2 | 3.31 | `mk-minchul/AdaFace` |
| SphereFaceR | iResNet100 | cosine | MS1 | 10 | `ydwen/opensphere` |

*Table 4.* Main characteristics of models used in this work.

# C. Macroscale analysis details

## C.1. Summaries: global average, inter entropy, intra entropy

The main quantity of interest is the intra-entropy, as defined in the main text. We have also computed the global average of each attribute and another entropic summary, the inter-entropy, which quantifies the variability of an attribute across identities. If $m_a^i = \frac{1}{|D_i|} \sum_{x \in D_i} a(x)$ is the attribute $a$'s average over identity $i$,

1. $\frac{1}{|I|} \sum_{i \in I} m_a^i$ is the attribute global average;
2. $H(m_a^i)$, the attribute inter-entropy is the differential entropy of a Gaussian kernel density estimation of $(m_a^i)_{i \in I}$;
3. $H_a$ is the attribute intra-entropy, as defined in the main text

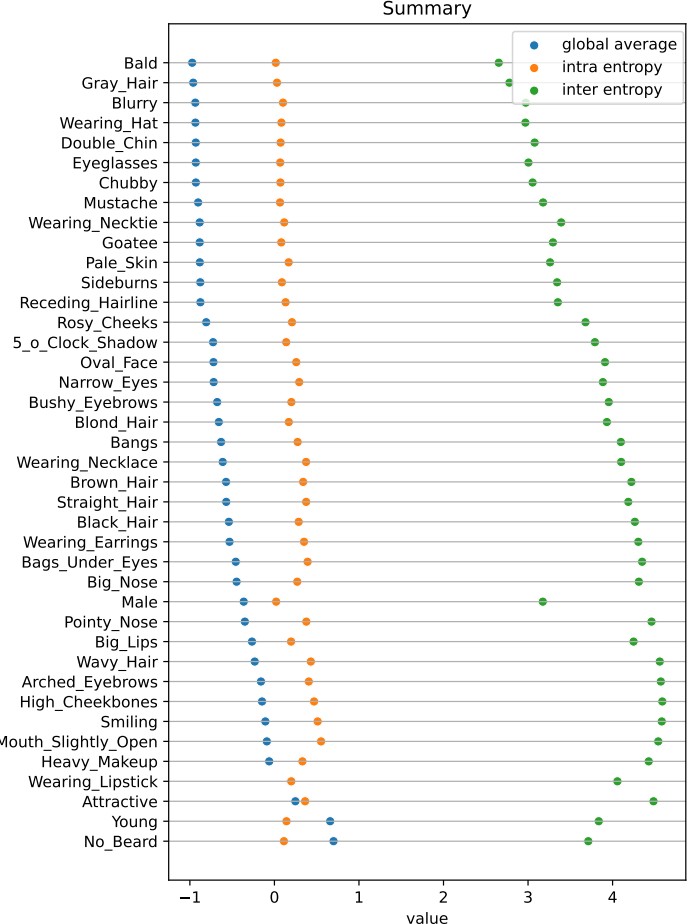

*Figure 6.* **CelebA attribute summaries.** The global average is measured on all images. The intra-entropy first computes the entropy at the identity level, then averages over all identities, and conversely for the inter-entropy.

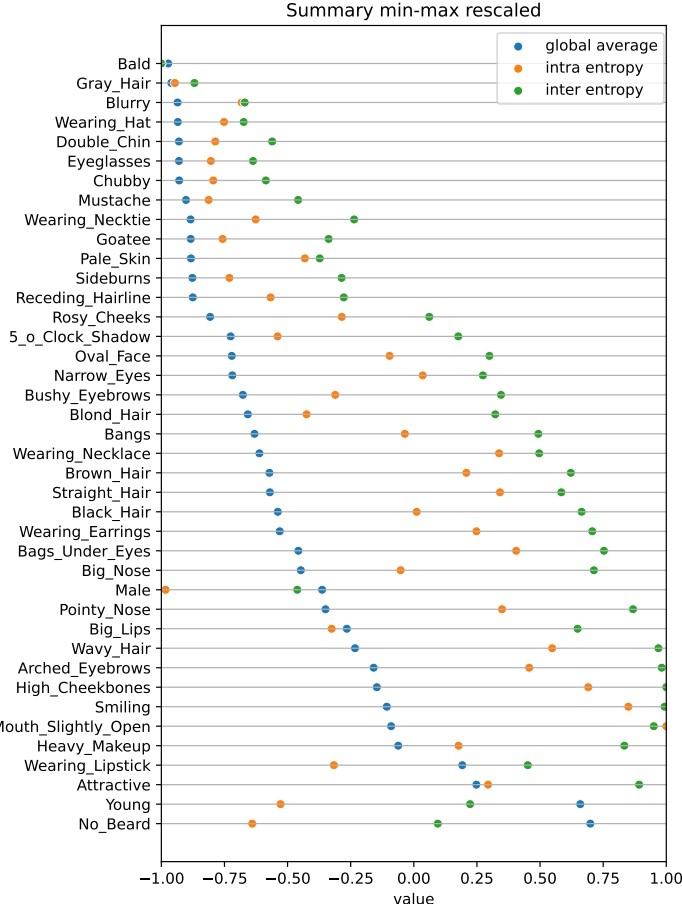

*Figure 7.* **CelebA attribute summaries, rescaled.** This figure is a min-max rescaled version of Figure 6 to enhance between summary comparison.

## C.2. Distributions of mean and intra entopy for each attribute

In Figure 8 and Figure 9 we show the estimated distributions of attribute mean and intra entropies for each attribute over our cleaned subset of CelebA.

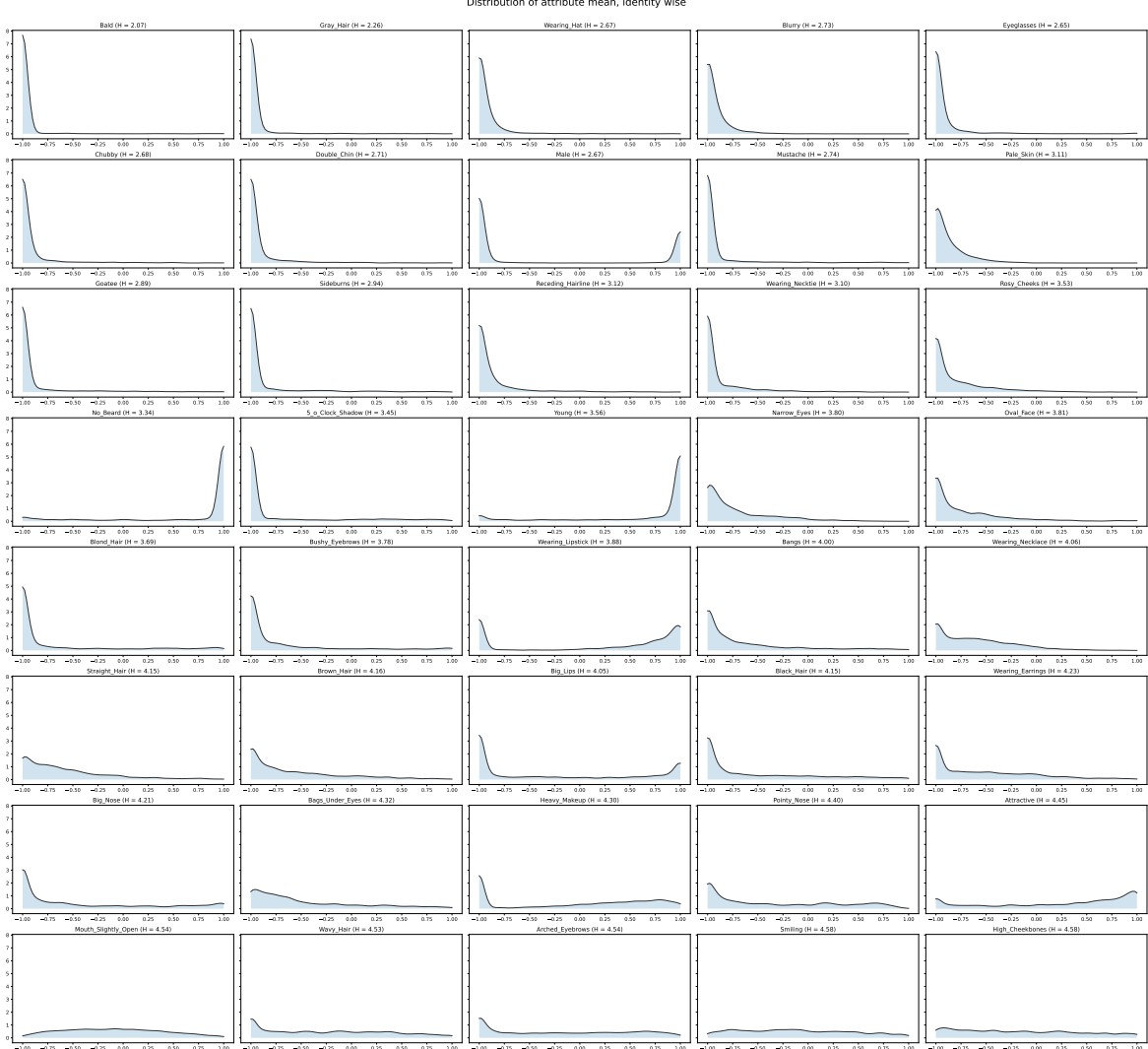

*Figure 8.* Distribution of attribute mean over identities of CelebA. In parenthesis is the entropy H of this distribution that correspond to the inter-entropy

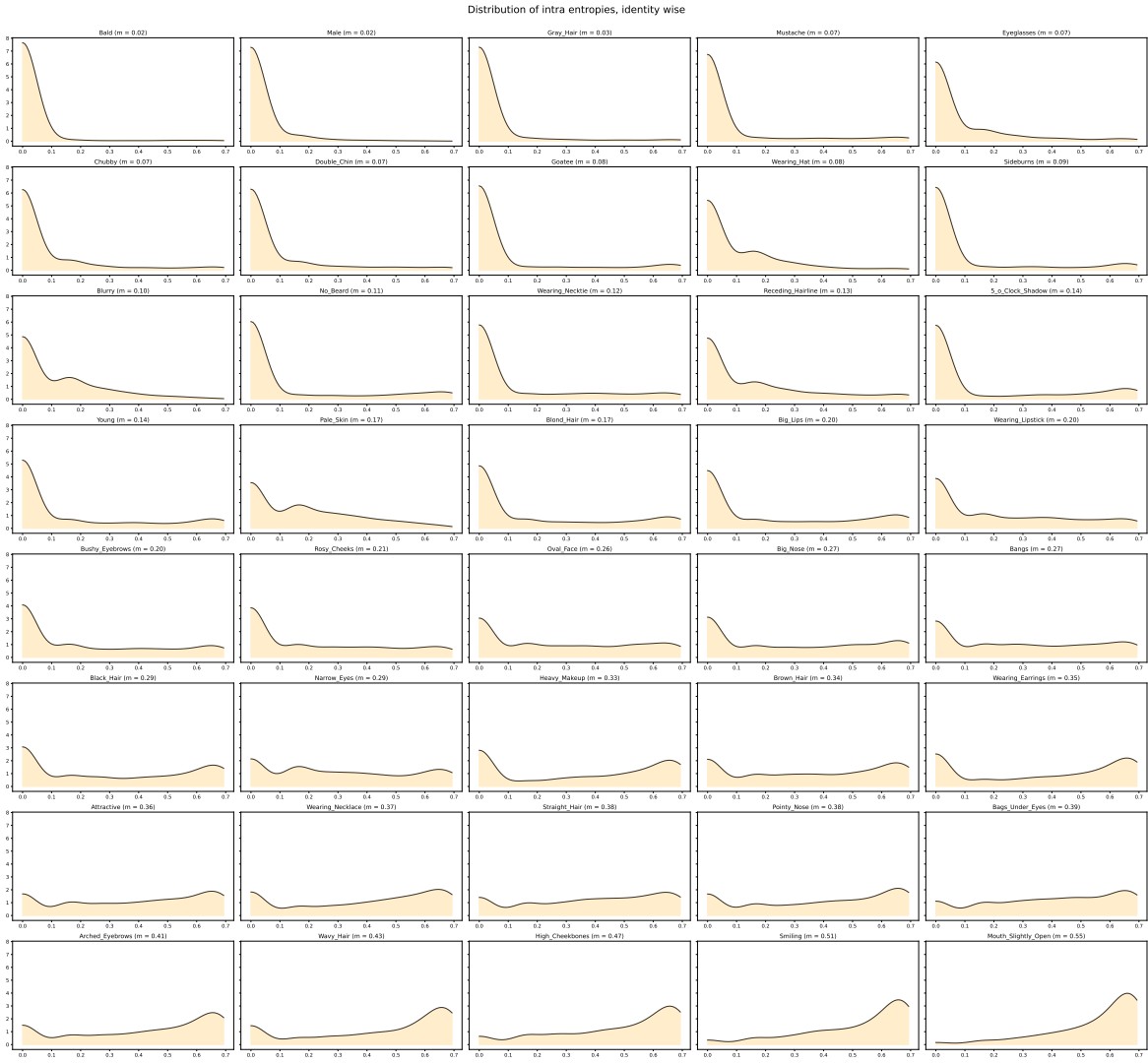

*Figure 9.* Distribution of intra entropies over identities of CelebA for each attribute. In parenthesis is the mean m of this distribution that correspond to the intra-entropy.

## C.3. p-values associated with the KS-test of $\mathcal{H}_0$

In Section 3.1 we perform a statistical test at the attributes-by-modality level, so in total $n_{attributes} \times n_{modalities} = 40 \times 2 = 80$ statistical tests for each models. All p-values are reported below, and while we don't want to make statements for single attributes, we can still notice that non-significant attribute-modality pairs are often the same for different models.

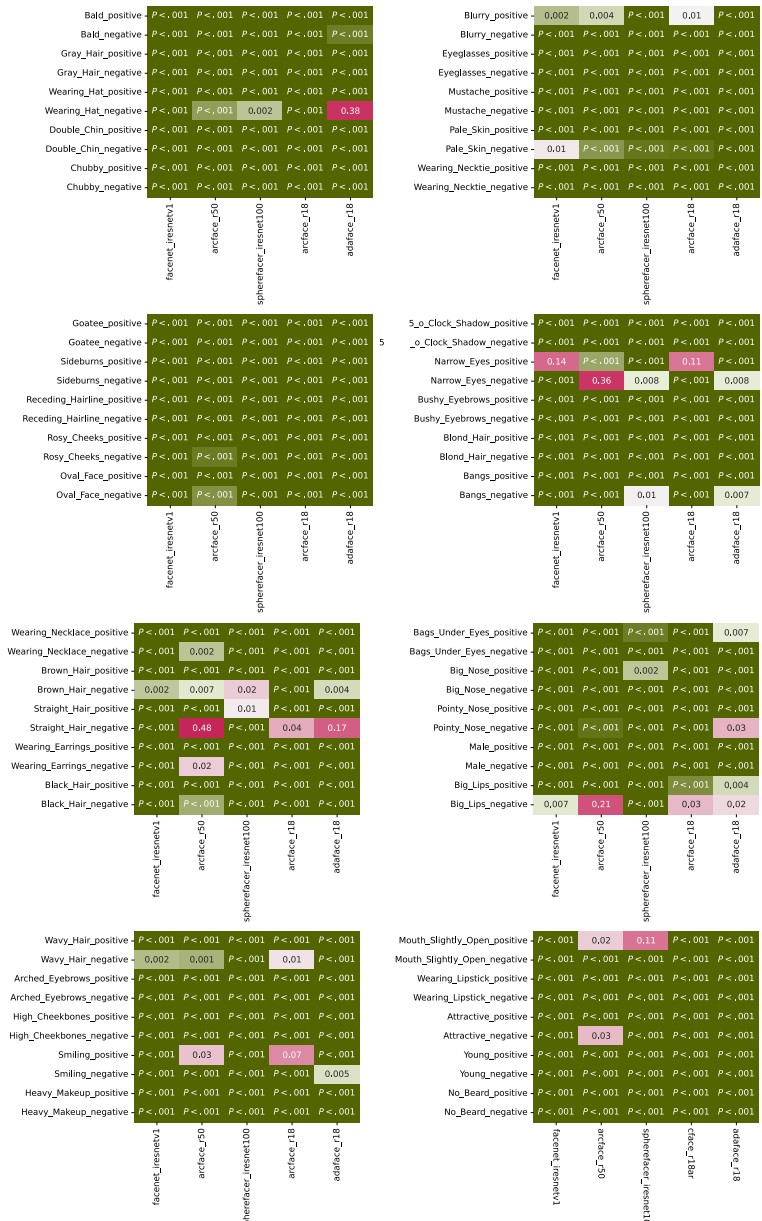

*Figure 10.* p-values associated with the KS test statistic for each $(attribute, modality)$ pair, for arcface, facenet, sphereface and adaface. We see how the the test result is significant for the majority of pairs.

# D. Image generation

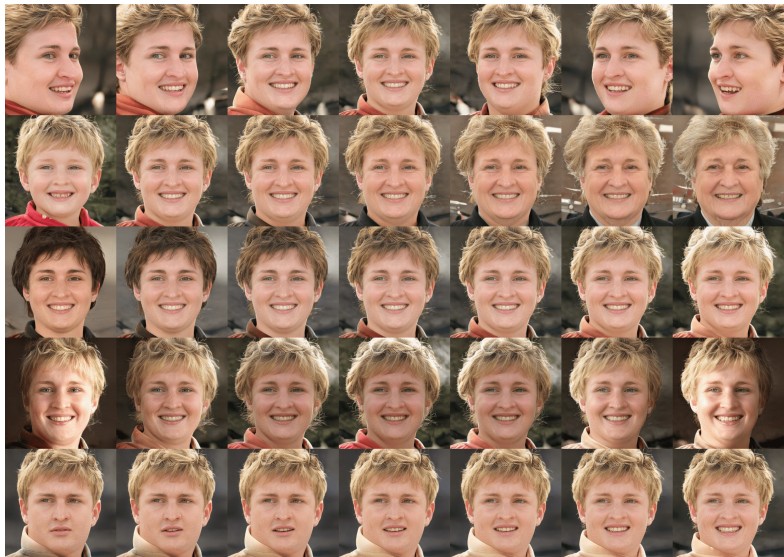

*Figure 11.* Example of variations produced with Gan-control, row-wise: orientation, age, hair color, illumination, expression. We use subsequences of the rows projected in embedding space to compute a vector field.

## D.1. Gan control

In theory, all that is needed for GAN-control is an attribute classifier to train the pipeline, but we use the pretrained model and hence have access to the following attributes: pose, age, hair color, illumination and expression. These attributes are associated with specific, non overlapping subspaces of the latent space of the GAN. In addition, there are two other subspaces in the latent space: identity and extra. The extra bandwidth stores everything neither related to the identity nor the attributes, like the background. The identity subspace parametrizes the identity. Each attribute can be modified by the user and can be multidimensional (e.g. expression has 64 dimensions) or not (e.g. age is a single number). Figure 11 presents generation samples.

## D.2. Low level augmentations

In addition to the complex variations generated by GAN-control, we compute simpler augmentations: brightness, hue and image quality. A summary of the 8 types of augmentations is presented in Table 5

| Attribute | Method | Start Value | End Value |
|---|---|---|---|
| Orientation (degrees) | GAN-control | -45 | 45 |
| Age (years) | GAN-control | 15 | 75 |
| Hair Color (RGB) | GAN-control | [52, 31, 10] | [215, 214, 124] |
| Illumination | GAN-control | Intense light from right | Intense light from left |
| Expression | GAN-control | Neutral | Wide Smile |
| Brightness | low level | -50 | +50 |
| Hue | low level | -4 | +4 |
| Image quality | low level | Blurring | Contrast enhancement |

*Table 5.* Summary of augmentations of attributes and their ranges.

## D.3. Generation details

We want to efficiently generate discrete curves of length $L = 3$ in $E$ along which a single attribute is varied, like in Figure 2. To do that we select a range of variation for each attribute and discretize it in on $K = 7$ values. Endpoints of this range

are indicated on Table 5. For computational efficiency we generate a lattice composed of a central hypercube spanning the range $\left[\frac{L-1}{2}, K - \frac{L-1}{2}\right]$ and extended in one dimension by $\frac{L-1}{2} = 1$ point on each side. We checked that augmented images don't move the embeddings far enough to change an identity by analyzing id retention plots such as the ones reported on Figure 12. The distance should be compared to a typical threshold value depending on the model (see Figure 5 for distances computed on LFW).

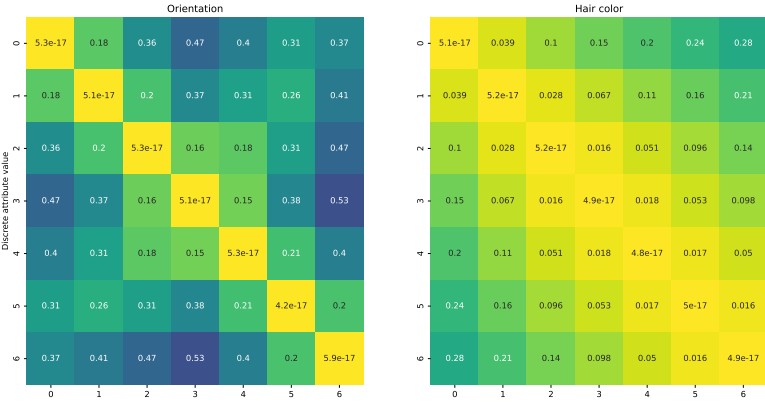

*Figure 12.* Id retention heatmap for ArcFace: each cell represents distances measured between different variations of orientation (left) and hair color (right). Identity is fixed.

# E. Energy experiments

## E.1. Details about the toy model

To obtain the toy model results of Figure 3, we trained 30 times a three layer MLP with hidden layer sizes 16, 16, respectively and 3-dimensional classification output. The training was performed using the Adam optimizer (Kingma & Ba, 2015) with learning rate $10^{-4}$, with a cross-entropy loss, on a dataset of 750 6-dimensional points divided in three classes as shown in the main text.

The attribute vector field for each dimension is computed by finite differences by perturbing the input dimension by $h = 0.05$ i.e., if $e = f(x_1, \ldots, x_6)$,

$$\tilde{v}_i(e) = f(x_1, \ldots, x_i + h, \ldots, x_6) - f(x_1, \ldots, x_i - h, \ldots, x_6),$$

then normalized to obtain unit-norm the vector fields $v_i$.

## E.2. Details about the FR models energy computation

We computed the energy for each FR model by sampling $10^4$ point from the 40 synthetic identity point clouds generated by GAN-control as it is described in Appendix D. For computational efficiency, fixing every point $e$, we compute the inner products between its vector and a set of 100 randomly sampled points at distance less than the chosen scale $\varepsilon$. The average distance for each point cloud is estimated by randomly sampling 10000 pairs of points.

The complete results are shown in Figure 13, integrated with the additonal SphereFace model (Liu et al., 2022) for comparison.

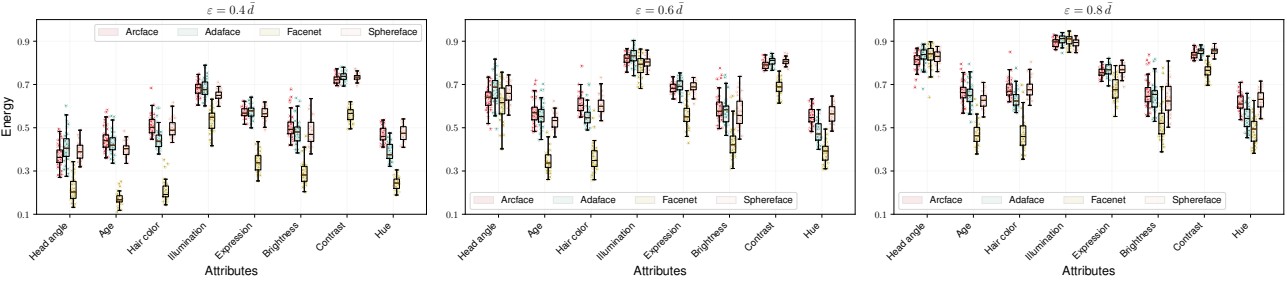

Figure 13. Energies for the pretrained FR models considered in the main text, with the additional Sphereface architecture (Liu et al., 2022).

# F. Fine-tuning details

## F.1. Datasets generation: train, validation and test

We generate similar train, validation and test sets of $10K$ identities each. Each identity is generated first from a *fixed image*, which is a unconditioned generated image with GAN-Control. For each attribute, we then generate variations of the fixed image with Gan-Control and low-level augmentations. For example, a fixed image that has an age value of $35$ will yield two other images with age values $45$ and $25$ and so on for other attributes for a total of $1 + 8 \times 2 = 17$ images by identity. When finetuning on an attribute $a$, we include in the train set only fixed images and their variations of $a$. ArcFace and AdaFace have parametric losses: for each train identity $i$, a parameter $c \in \mathbb{R}^p$ is learned during training to represent a centroid of $P_i$, which allows to compute the loss in a pointwise manner. We place ourselves in the open-set scenario, therefore our validation set contains (exclusively) identities not seen during training and hence we cannot directly measure the loss. To adapt this idea on hold-out data, we choose fixed images as parameters, making the loss parameter-free. Let $I_1$ and $I_2$ be validation identities with fixed images $x_1^*$ and $x_2^*$ and augmented images $x_1^+$ and $x_1^-$. Then this parameter-free loss decreases with $d(f(x_1^*), f(x_1^+))$ and $d(f(x_1^*), f(x_1^-))$. It increases when $d(f(x_1^*), f(x_2^*))$ decreases.

## F.2. Learning curves

We give details on our fine-tuning strategy and monitoring.

### F.2.1. LOSS CURVES

We present a sample of learning curves during fine-tuning on Figure 14 and Figure 15. In each plot, an attribute is chosen and the model is fine-tuned on it. Note the agreement between the attribute fine-tuned and the loss measured datasets containing different types of augmentations.

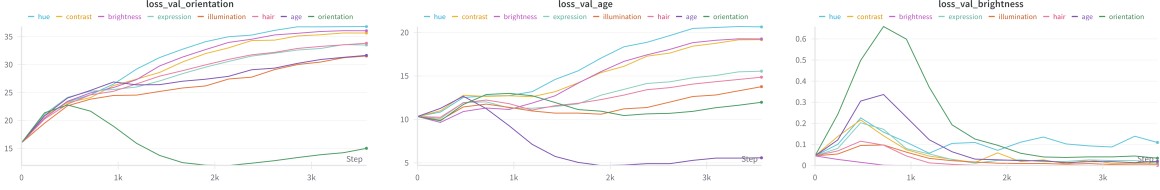

*Figure 14.* Validation losses for orientation, age and brightness finetunings of ArcFace

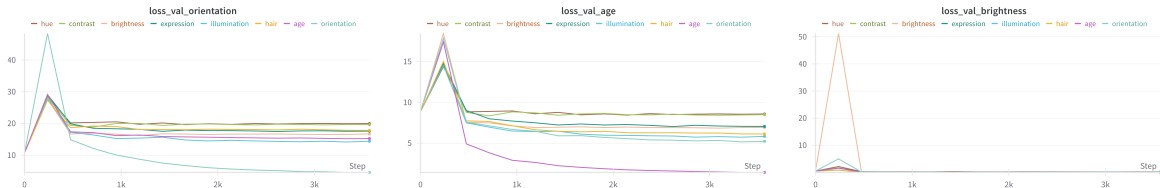

*Figure 15.* Validation losses for orientation, age and brightness finetunings of AdaFace

### F.2.2. ROC CURVES

We also track ROC curves along fine-tuning, both on generated data and on CelebA.

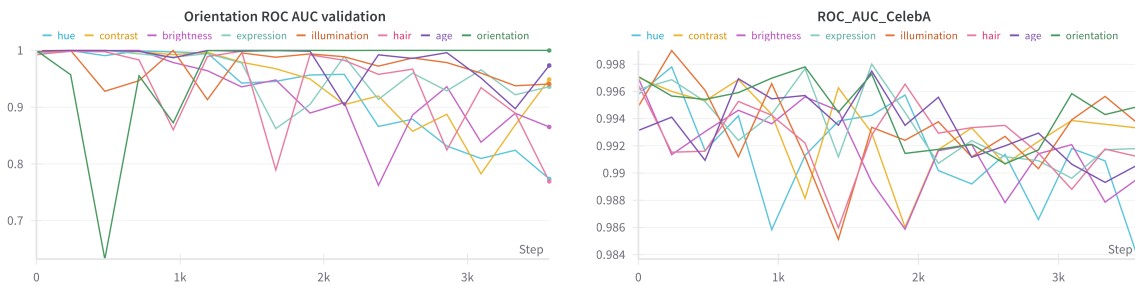

*Figure 16.* **Left** ROC AUC curve for ArcFace resnet 18 measured on a monovariant dataset of orientation variations. We see that the model fine-tuned on orientation performs the better. **Right** Along finetuning performance on CelebA decreases, but stays at a decent level.

### F.2.3. DISTANCE CURVES

Finally, we follow distances in embedding space along fine-tuning.

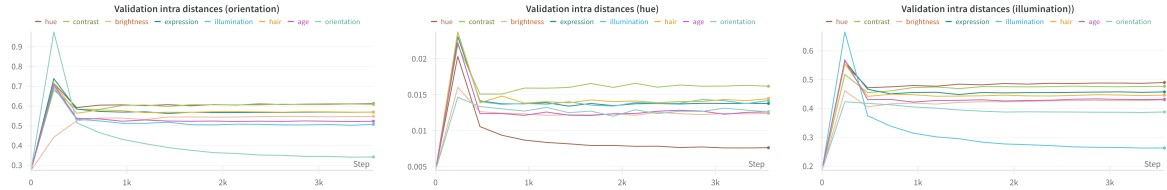

*Figure 17.* Evolution of distances for specific variations during finetuning of AdaFace resnet 18. We see that the model fine-tuned on the corresponding attribute has the smallest distances.

### F.2.4. TEST METRICS

Evaluation of the best model checkpoint on the test set yields the following sanity check results:

1. A model fine-tuned on attribute $a$ has a ROC AUC test equal to or extremely close to $1$ when evaluated on images subject to variations of $a$.

2. A model fine-tuned on attribute $a$ has the lowest test parameter-free loss of all fine-tuned models when evaluated on images subject to variations of $a$.

### F.3. Hyperparameters

For all finetunings, we always start from the same baseline, which is an ArcFace resnet 18 model trained on MS1MV3 (Deng et al., 2019a) for the first set of finetunings, and an AdaFace resnet 18 trained on VGGFace2(Cao et al., 2018) for the second one. We use ArcFace as a reference and when possible match end-of-training configuration (the training done by the providers of the model). We train both models for a maximum of $15$ epochs and we use callbacks on the validation loss to get the best checkpoint. Both model have a batch size of $42$ identities, presenting always all images of an identity in the same batch. We let both models have the same optimizer parameters: a small learning rate at $10^{-4}$ weight decay at $5 \times 10^{-4}$, momentum equal to $0.9$. For ArcFace we use the default $0.5$ rad for the margin while we set the learning rate of the loss to $10^{-3}$. For AdaFace, we let the default margin variability parameter, $m$ to $0.4$, the concentration parameter, $h$ to $=0.333$, the scale parameter, $s$ to $64$, the running batch mean coefficient to $0.01$ and we set the learning rate of the loss to $10$.

### F.4. Pipeline diagram

We show a schematic representation of the microscale experiment on Figure 18.

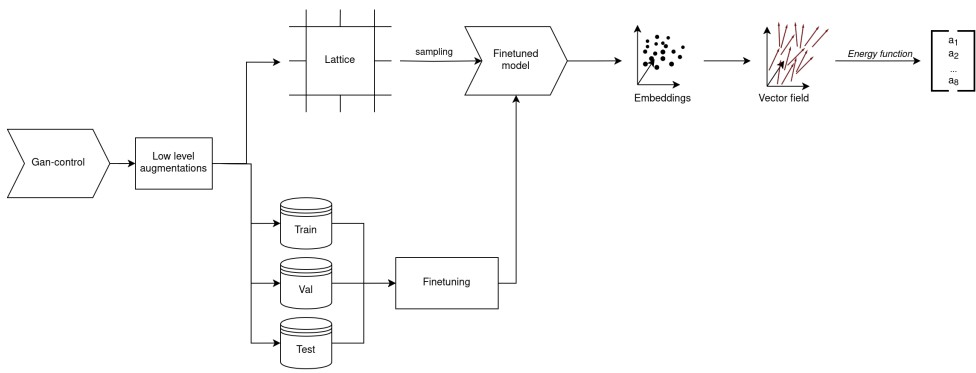

*Figure 18.* Complete pipeline of the microscale experiment.

## F.5. Fine-tuned energies

We show the complete set of energies for all attributes, scales and fine-tunings in Figure 19.

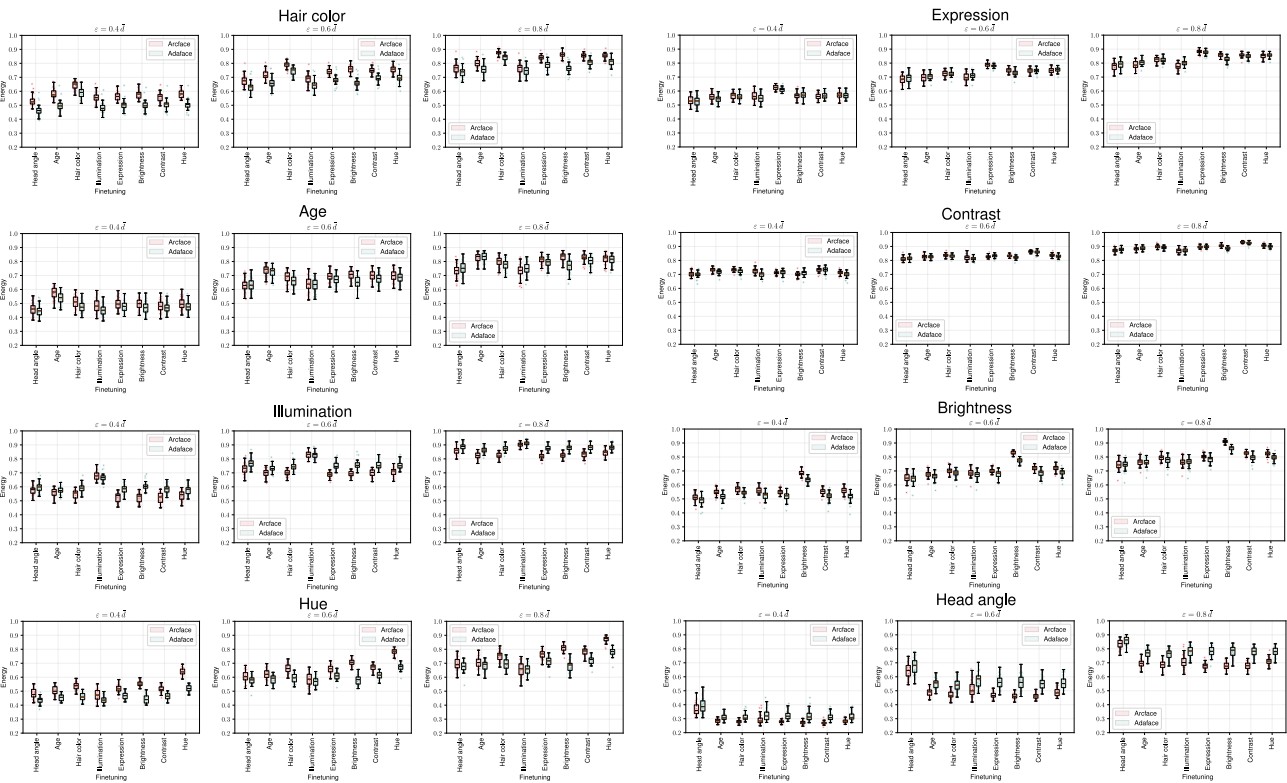

*Figure 19.* Energy results for each attribute and each fine-tuning of ArcFace and AdaFace.

