# OpenReview forum: "Attributes Shape the Embedding Space of Face Recognition Models"
_ICML.cc/2025/Conference — ICML 2025 poster_

### Official Review · Reviewer_43PU · 2025-02-20

**Overall Recommendation:** 2

**Summary:**

The paper studies the organization of embeddings in face recognition task with respect to facial attributes. The study is mainly focussed on ArcFace and FaceNet models, while the study uses results on LFW and CelebA.

## Update after rebuttal
In my initial review, most of the local issues raised were w.r.t clarity. They have been answered and while I cannot pinpoint to one as being wrong or as having not addressed completely the issues, the answers have raised the clarity only partially. The explanation are ample, with many mathematical formulation that need to be matched with the larger explanation from the paper. Thus the paper needs to be rebuild with all new information in its place to have a better understanding.

Overall, I have raised my grade from 1 to 2, but am I not confident to raise it in the "acceptance range" because:
 - while authors have provided their arguments that FR theme is treated in NIPS and ICML, I maintain that this paper is better suited in a "Face" or "Biometric" conference. The mentioned papers are scarce and distributed over several years. In my view, a paper  would need to be very strong on a niche theme attract auditorium in ICML.
 - the clarity is improved. But now there are too many pieces on the table. I feel that I need to reread a complete paper to be certain that the paper is clear. I am still not convinced that the energy has value for a reader.

**Claims And Evidence:**

The paper makes relatively vague claims. They can be inferred from page 2 l88-l108, left column. They would be:
 1. "a procedure to check if attributes shaping the relations inter-identities are the ones most deterministically linked to an identity"
 2. "an invariance energy measure to quantify the invariance of the embedding model to each attribute"

A problem with both is that steps are taken towards achieving them, but the results are not clear.

For claim 1, this is treated in section 3.2 but the results are in appendix B (outside the paper and therefore not mandatory to be read for  review). In Appendix B (where space is unlimited) we find  figure 6 (where writing and all graphical signs are too small) and without explanation. What is better? What is worse? In section 3.2 it is mentioned that there are two models yet in the appendix there is a single set of results. Some results are showed again in figure 1d, but that is again too small and it is not quite clear what is there (w.r.t what is expected).

For claim 2, the paper indeed proposes an energy, but the derivation results and interpretation are not very appealing. Following section 4, it is not clear to me if "lager is better" or viceversa.

**Essential References Not Discussed:**

I believe the paper is fine on this criteria.

**Ethical Review Concerns:**

The paper approaches the face recognition theme, which is a sensitive one, but in my view, the approach, the proposal and the finding do not raise any ethical flag.

**Experimental Designs Or Analyses:**

The experimental design is arguable. While the benchmark and the problem make sense, the assumed methodology, results presentation and explanation provided are not fluent. The proposed energy is not something that reader can take away and used it to explore their models.

**Methods And Evaluation Criteria:**

The benchmarks make sense. The metrics make less sense. Alternatively, the paper might revisit the explanation and make it more clear.

**Other Comments Or Suggestions:**

I have not notice typos.

**Other Strengths And Weaknesses:**

The paper approaches an interesting direction that definitely is worth of investigation. My concerns are the following:
 1. instead of  ICML the paper might be better suited to "Face" or "Biometrics" dedicated conference. The significance of the results  is less relevant for the general Machine Learning community and more for the face recognition.
 2. the paper is not clear. The method used to investigate the relevance of attributes and their impact over the mapping  providing embeddings lack clarity. On the formal aspect, the paper sends the reader too many times to inform in the appendices, but some details are omitted there, too. Figures are too small and it is not explained what should be seen. On the information, the paper chooses to use a mathematical formulation, but often fails to clarify things. For instance why do we need to define a space of a transform  of  a specific attribute, which is than simulated with a GAN?
3. The findings are not clear. There are conclusions that try to summarize the findings, but they are vague and poorly correlated with the results (or with results presentation?)

**Questions For Authors:**

- What is meant by "macroscale" and respectively by "microscale"?

- What distance (i.e. L2, Cosinus, etc) is used for Table 1? And the pretraining process was done on which database? This may be related to information presented in Appendix A ("The models where the embedding space is equipped with cosine distance (AdaFace, ArcFace, SphereFaceR) show the distance in degrees") but a plain explanation would be better.

- Similar questions for table 2. The caption and text explanation make things very vague ("we find significant negative Spearman correlations between the intra-entropies for each attribute and the previously obtained KSa (Table 2)")

- Eq (2) what is meant by "\circle" between f and \alpha?

**Relation To Broader Scientific Literature:**

The paper approaches a deconstruction of the model used to provide embeddings in the face recognition tasks. There are previous results within the same trend, but the precise direction of the paper is not, to my best knowledge, previously explored.

On other hand, the specific of the paper, the findings... I am not sure that ICML is the best place to present them; they seem to be more suitable in a Face dedicated conference (IEEE Face&Gesture, Biometrics, etc.)

**Theoretical Claims:**

The paper does not contribute significantly on theoretical side.

---

> ### Author Rebuttal · Authors · 2025-03-31
>
> Here $E$=embedding space; $A$=attribute/s; $MS$=macroscale; $ms$=microscale
>
> 1.*"Figures too small, unreadable"*
>
> Thank you for the comment.
> Anonymous repo shows improved Figs and Tables: https://shorturl.at/12ZFg
>
> 2.*"Paper uses ArcFace and FaceNet while uses results on LFW and CelebA. Meaning of $MS$ and $ms$"*
>
> We point to the summaries provided by the 3 other reviewers.
> In brief, we study how interpretable $A$s of images impact the geometry of the $E$ of FR models.
> We clarify that we use LFW to compare the intra or inter-class distances ($d_b$, $\overline{d}$) for various models (Table 1).
> This motivates us to study $E$ through the lenses of $A$s at two scales (lines 137-143): $ms$ and $MS$, focusing on intra or inter-class geometry.
> The $ms$ relates to single identity point clouds (images as points); $MS$ to the whole $E$, (identities as points).
> We use CelebA to analyze $MS$ as it is endowed with interpretable $A$s.
> We explore the $ms$ through mapping augmented data on $E$ according to $A$s given by the latent space of GAN-Control and other $A$s (brightness, contrast, hue).
>
> 3.*"Vague claim 1; unclear Fig 1d. The paper sends the reader to Appendix. Vague Table 2"*
>
> We agree to improve Sec. 3 discussing $MS$ (see anon. repo) and the Appendix.
> However, we clarify that the results are in the main paper: Table 2 and Fig. 1d.
> Due to space constraints, Sec. 3 shows:
> (a) How to compare the $E$ of ArcFace and FaceNet through KSdist from the lenses of binary interpretable $A$s.
> (b) How FaceNet differs from ArcFace since the KSdist are higher (see new Fig. 1d, scatterplot higher than the diagonal), showing that for FaceNet, the $E$ has higher structural dependency on the $A$s.
> (c) Significant negative correlation (Table 2) between the intra-entropy of an $A$ (= *how much the $A$ varies inside each identity point cloud*) and the KSstat (= *how much an $A$ shapes the global geometry of $E$*), confirmed by the "bubble-size" of the scatterplot of Fig. 1d: the more the $A$s vary within identities, the less they shape $E$ at $MS$.
>
> Fig.6 analyzes only the CelebA $A$s, no mapping on $E$: the $A$s with greatest (bald, male) or lowest (mouth-slightly-open, smiling) intra-entropy from new Fig. 6b have high and low KSstat on Fig. 1d, respectively.
>
> 4.*"Claim 2: derivation and interpretation of the energy not appealing. From Sec. 4, unclear if "lager is better"*
>
> We refer to Sec. 2, then Sec. 4.1 and Fig. 2c.
> Sec. 2 observes that FR models achieving better identity recognition should be approximately invariant to $A$s.
> Thus, in Sec. 4.1, we conjecture that an FR model has high sensitivity to an $A$ if the associated local vector-field derived from the data augmentation on the same $A$ has low energy (Fig. 2c "Aligned" $\mathcal{E}=0.0$ compared to $\mathcal{E}=0.75$ "Unaligned").
> We conclude that the model is more invariant to $A$s if the energy is higher ("higher energy is better").
>
> 5.*"Energy is not something that one can take away to explore their models."*
> Our methods rely on either (1) an attributed dataset or (2) controllable data augmentation.
> If one wishes to diagnose how a particular $A$ is treated by a model, one needs to measure and/or control $A$.
> Since metadata labeling is expensive, controllable augmentation is the easiest way.
> As noted by Reviewer EQbt, one of the energy usages is to guide the fine-tuning process to achieve a better FR model.
> The energy measures the $A$s over which an FR model is less invariant.
> Thus, it not only shows the model's limitations but is also a quantitative tool for investigating potential biases and providing directions for model improvement and resilience.
>
> 6.*"Paper might be better suited to FR conference. Results less relevant for the general ML community."*
>
> We note that our references encompass previous works covering solely the FR domain and appearing in general ML conferences:
> (a) ICML2021 "Larnet: Lie algebra residual network for FR";
> (b) NeurIPS2024 "TopoFR: A closer look at topology alignment on FR".
> Inspired by those, we suggest that FR is a significant domain for general ML development, specifically for representation learning.
> Indeed, it deals with a high-dimensional domain, where the low-dimensional manifold hypothesis has been verified (e.g., "The intrinsic dimension of images and its impact on learning." ICLR2021), so that embedding projection and metric learning are meaningful approaches.
> In addition, it motivates us to interpret $E$ because of the sensitive nature of identity recognition for face images.
> Furthermore, past works have driven a valid effort to collect images with metadata, thus enabling further analysis of $E$.
> Moreover, the hierarchical distinction of $MS$ and $ms$ can be extended to other open-set metric learning frameworks.
>
> 7.*"Distance for Table 1? Pretraining on which database?*
>
> See Table 5 in anon. repo listing the used models.
> FaceNet with Euclidean, others with cosine.
>
> 8.*"Eq2 circle"*
>
> Composition of functions.

---

### Official Review · Reviewer_MCoe · 2025-03-14

**Overall Recommendation:** 3

**Summary:**

This paper provides comprehensive analysis of the embedding space of face recognition models. Through a geometric structure perspective, this work analyses the macroscale and microscale structures of the embedding space, and quantifies how human interpretable attributes influence the structures. Experimental results show that the proposed "invariance measure" can well quantify the sensitivity of FR models to different attributes, and targeted finetuning strategy guided by the "invariance measure" effectively brings performance gains on targeted attributes.

## update after rebuttal
The authors have provided detailed and reasonable responses. Though it would be better if constraints of the proposed method could be relaxed and more interesting analysis could be observed, this paper in current form is ok for ICML. So, I would like to keep "Weak accept" rating unchanged.

**Claims And Evidence:**

N/A

**Essential References Not Discussed:**

N/A

**Experimental Designs Or Analyses:**

N/A

**Methods And Evaluation Criteria:**

N/A

**Other Comments Or Suggestions:**

N/A

**Other Strengths And Weaknesses:**

[Strengths]
- This work provides deeper and interesting interpretability of FR embeddings from a geometric structure perspective.
- The proposed "invariance measure" can well quantify the sensitivity of FR models to different attributes,  so that  can suggest new avenues for improving the robustness of FR systems.
- This paper is well written and easy to follow.

[Weaknesses]
- The calculation of "invariance measure" relies on strictly controlable face images, so that it can only be calculated and analysed on GAN-generated images currently, which may limit its' real-world applications.
- Though sounds reasonable, the observations in Figure 4 are not surprising. Analysis of sensitivity of face recognition models with respect to attributes can also be accomplished by constructing face recognition testing sets featured with specific face attributes. Some "non-trivial" observations through the proposed "invariance measure" may help highlight the contribution of this work.

**Questions For Authors:**

N/A

**Relation To Broader Scientific Literature:**

N/A

**Theoretical Claims:**

N/A

---

> ### Author Rebuttal · Authors · 2025-03-31
>
> 1. *"The calculation of 'invariance measure' relies on strictly controlable face images, so that it can only be calculated and analysed on GAN-generated images currently, which may limit its' real-world applications."*
>
> Our work requires the ability to act meaningfully on the input space, which can be challenging. However, while GANs provide a powerful tool for generating controlled variations in complex attributes, they are not strictly necessary. Low-level data augmentation techniques, such as rotations or brightness adjustments, can also be used effectively to create variations in input data. The brightness, contrast, and hue in Fig. 4 belong to this class of attributes. Moreover, our proposed methodology is not limited to image data. It can be applied to interpretable inputs, such as tabular data (cf. our toy model validation experiment, where we don't use a GAN). In this case, sensitivity analysis methods are well-established, but our approach, specifically aimed at measuring invariances for a set of continuous attributes, could be complementary.
> Finally, transforming complex input data meaningfully remains challenging, but architectures like GAN-Control show that the ability to predict is enough to gain some controllability. Rapid advances in generative modeling may further enhance the applicability of our approach.
>
> 2. *"Unsurprising results for the invariance energy."*
>
> In Fig. 4, panels c and d are about the fine-tuning validation, so we assume your comment refers to panels a and b. The results shown in panels a and b align with expectations and, in fact, mirror findings from the macroscale experiment: they show that interpretable attributes have a greater influence on FaceNet's embedding space compared to ArcFace or AdaFace.
> However, the point we want to highlight is the methodology, which is different. The invariance measure allows us to analyze continuous data, for which scales are incomparable in a principled manner. For instance, it's challenging to directly compare the scale of age with that of hair color. By focusing solely on directional information, our approach provides coherent results without assuming a relation between data scales, but while, concretely, both macro- and microscale experiments use testing image sets, the tools to analyze them differ.
>
>
> 3. Unrelatedly, here we provide a link to an anonymous repository https://shorturl.at/12ZFg to share updated figures and tables following the comments of other reviewers

---

> > ### Comment · Reviewer_MCoe · 2025-04-05
> >
> > Thank you for your detailed responses. Most of my concerns have been addressed.

---

### Official Review · Reviewer_zixr · 2025-03-15

**Overall Recommendation:** 3

**Summary:**

This paper investigates the geometric structure of the embedding space in Face Recognition (FR) models, focusing on how human-interpretable facial and image attributes influence the learned representations. FR models, which use deep learning and contrastive losses, aim to map images of the same identity closer together in a high-dimensional space. However, the learned embeddings also encode other attributes such as hair color or image contrast, which can impact model performance and fairness. The study introduces a new physics-inspired alignment metric to analyze the dependence or invariance of FR models to these attributes.

**Claims And Evidence:**

Yes.

**Essential References Not Discussed:**

None

**Experimental Designs Or Analyses:**

Yes. Please see "Methods And Evaluation Criteria".

**Methods And Evaluation Criteria:**

The paper presents a comprehensive methodology for analyzing the geometric structure of the embedding space in Face Recognition (FR) models, with a focus on attribute invariance. However, there are several areas where both the methods and experiments could be improved or expanded:

1. The invariance energy measure is primarily evaluated on the CelebA dataset, which, while popular, may not encompass the full range of variability seen in real-world face recognition tasks. The dataset is limited in terms of facial attributes and may not capture all the complexities of face recognition across different demographics, environments, and conditions.

2. While the study emphasizes the geometric structure of the embedding space, it could further enhance the interpretability of the results by visualizing the learned embeddings in a more user-friendly way.

3. The study compares a few models (FaceNet, ArcFace, AdaFace), but it would benefit from a more thorough cross-model analysis.

4. While the study focuses on fine-tuning with single attribute augmentation, it would be better to fully explore the effect of multi-attribute data augmentation on the embedding space.

5. Although the paper provides some insights into the geometric structure of the embedding space, it could be difficult for non-experts to intuitively understand how the changes in embedding space occur due to different attributes.

**Other Comments Or Suggestions:**

none

**Other Strengths And Weaknesses:**

Strengths:
1. The writing of this paper is good, and the structure is easy to follow.

2. The proposed method sounds reasonable.

3. The experimental results look good.

Weaknesses:

1. The invariance energy measure is primarily evaluated on the CelebA dataset, which, while popular, may not encompass the full range of variability seen in real-world face recognition tasks. The dataset is limited in terms of facial attributes and may not capture all the complexities of face recognition across different demographics, environments, and conditions.

2. While the study emphasizes the geometric structure of the embedding space, it could further enhance the interpretability of the results by visualizing the learned embeddings in a more user-friendly way.

3. The study compares a few models (FaceNet, ArcFace, AdaFace), but it would benefit from a more thorough cross-model analysis.

4. While the study focuses on fine-tuning with single attribute augmentation, it would be better to fully explore the effect of multi-attribute data augmentation on the embedding space.

5. Although the paper provides some insights into the geometric structure of the embedding space, it could be difficult for non-experts to intuitively understand how the changes in embedding space occur due to different attributes.

**Questions For Authors:**

none

**Relation To Broader Scientific Literature:**

/

**Theoretical Claims:**

Yes.

---

> ### Author Rebuttal · Authors · 2025-03-31
>
> 1. *"The invariance energy measure is primarily evaluated on the CelebA dataset, which, while popular, may not encompass the full range of variability seen in real-world face recognition tasks."*
>
> We want to clarify that we use CelebA for the macroscale experiment to compute the KS statistic on a real-world dataset, but GAN-Control is our main source of images for the invariance energy experiments.
> GAN-Control allows us to create a large number of variations of an image in a systematic and scalable way, which would not be possible with real images.
> In particular, we obtained more than 120K images by identity, by varying over 8 continuous latent variables corresponding to the 8 attributes. We put the details in Appendix C.2 of the submitted paper.
> Still, we acknowledge that the generated variability for each attribute is limited by the GAN-Control quality of generation.
>
> 2. *"the study ... could further enhance the interpretability of the results by visualizing the learned embeddings in a more user-friendly way."*
>
> The high-dimensional nature of embedding spaces in state-of-the-art models inherently limits the fidelity of visualizations. However, for the microscale analysis, we believe that visualizing the vector field over the embedding space, rather than just the embeddings themselves, can enhance interpretability.
> In Figure 2c, we illustrate this concept using a toy model in a low-dimensional embedding space. Although this simplification doesn't fully capture the complexity of high-dimensional face recognition models, it provides a conceptual framework. For instance, an attribute like *contrast* might appear more "disordered" compared to "head angle," aiding readers in understanding the underlying measure.
>
> 3. *"The study compares a few models."*
>
> We kindly refer to the answer 4 given to reviewer EQbt, who raised a similar point.
>
> 4. *"multi-attribute data augmentation on the embedding space."*
>
> When creating synthetic identity point clouds, we augment the starting image with all combinations of attributes (see Appendix C.2 for details). However, when computing the invariance measure, we sample curves with "infinitesimal" variations along a single attribute to approximate the natural definition of the vector field.
> While considering multiple attributes simultaneously is intriguing, it is unclear what geometric construct would best represent this scenario. One could explore the energy of a k-vector field or average the vector fields across different attribute augmentations. Although this presents an interesting research direction, we defer this investigation to future work.
>
> 5. *"Intuitive understanding of the change in embedding space."*
>
> Visualizing changes in high-dimensional spaces can be challenging, even for experts. Our goal in this work is to provide quantitative insights on how embeddings behave when slightly varying an interpretable attribute.
> If modifying an attribute shifts the embeddings in a "random" way, the energy will be high. Conversely, an attribute modification yielding very predictable changes in embeddings would correspond to an ordered embedding space.

---

### Official Review · Reviewer_EQbt · 2025-03-17

**Overall Recommendation:** 4

**Summary:**

The paper describes a multi-scale geometric structure in embedding space created by Face Recognition (FR) models' feature embedding. The paper proposes a geometric-based approach to understand the influence of facial and image attributes to FR models. A physics-inspired alignment metric is also introduce.
The main findings help understand the models having some degrees of invariance across various attributes. This leading to deeper interpretability of the models' strengths and weaknesses.

**Claims And Evidence:**

Yes

**Essential References Not Discussed:**

None

**Experimental Designs Or Analyses:**

Yes. The Macroscale and microscope analysis make sense as they're looking at different scale with regards of identity, global (across multiple identities) vs. local (within one identity).

**Methods And Evaluation Criteria:**

Yes

**Other Comments Or Suggestions:**

There should be a section to conclude on the influence of attributes to both macroscale and microscale with specific examples and further analysis as suggested in the weaknesses section.

**Other Strengths And Weaknesses:**

One potential strength is the invariance energy measure is that it could help guide the fine-tuning process to achieve a better FR model.
However, the paper has some weaknesses/limitation as follows.
1. Lacking of some further analysis on some attributes overlapping between macroscale and microscale analysis, e.g. hair color, age, expression.
2. Missing a reference to a sub-figure in the paper, i.e. Fig. 1c.

**Questions For Authors:**

1. Why only two FR models are chosen for the macroscale and microscale analysis in the paper? How about the other models shown in table 1?
2. What is the architecture of the ArcFace used in the analysis? Is it ResNet18 or ResNet50?

**Relation To Broader Scientific Literature:**

The key contributions of the paper provide some basic and metrics to be used to understand any issue of the pre-trained FR models and how fine-tuning may help to improve them.

**Theoretical Claims:**

There is no theoretical claims and proofs given in the paper. The paper defines metrics and formulas to obtain numbers for experimental analysis.

---

> ### Author Rebuttal · Authors · 2025-03-31
>
> 1. *"There should be a section to conclude on the influence of attributes to both macroscale and microscale with specific examples and further analysis as suggested in the weaknesses section... Lacking of some further analysis on some attributes overlapping between macroscale and microscale analysis, e.g., hair color, age, expression."*
>
> Thank you for the interesting comment. We don't have a detailed answer for the current work as the data type is different for the macroscale and microscale analysis (binary VS continuous); in addition, for the macroscale the attributes come from CelebA metadata, whereas for the microscale, the attributes are derived from GAN-Control; therefore, we think the mapping from both experiments is not straightforward. In general, however, the attributes in the micro and macro-scale are explored in different ways, even if they share the same name.  The microscale analysis describes the effect of *small* variations of images of the same identity, while the macroscale describes the effect of *large* variations for images belonging to different identities.
>
> 2. *"Missing reference to Fig. 1c."*
>
> Thank you for the suggestion. We will include the missing reference in Sec. 3.1.
>
> 3. *"One potential strength of the invariance energy measure is that it could help guide the fine-tuning process to achieve a better FR model."*
>
> We thank the reviewer for the comment. Indeed, one of the possibilities of making use of the proposed microscale invariance energy measure is to improve the recognition performance after measuring the invariance energy on meaningful attributes. In particular, as suggested, we can think of using the invariance measure to drive further fine-tuning on specific attributes.
>
> 4. *"Why only two FR models are chosen for the macroscale and microscale analysis in the paper? How about the other models shown in table 1?"*
>
> For the macroscale experiment, we filled the gap and conducted the experiment with all models mentioned, as it is computationally feasible.
> Results are reported in the updated Table 2 on this anonymized repository: https://shorturl.at/12ZFg.
> The conclusions remain consistent: the embedding spaces' macroscale structures are most significantly influenced by attributes with the lowest intra-entropy, i.e., those most deterministically linked to an identity.
> For the microscale experiment, we computed the energy for four models (AdaFace, FaceNet, SphereFaceR, ArcFace-ResNet18) as shown in Figure 12 of the Appendix. However, due to computational constraints, we performed fine-tuning validation only on two of the best-performing models.
> A comprehensive analysis of all combinations of losses and backbone architectures for the FR models is beyond the scope of this paper.
> We believe the results discussed are sufficient to demonstrate that the proposed microscale and macroscale analyses are significant and valuable for exploring the embedding space through interpretable attributes.
>
> 5. *"What is the architecture of the ArcFace used in the analysis? Is it ResNet18 or ResNet50?"*
>
> For the macroscale, table 2 of the paper originally reported results for ArcFace with a ResNet50 backbone. The updated table 2 in the anonymized repository (same link as previous answer) now reports results for both backbones and other models. For the microscale analysis, AdaFace and ArcFace have the ResNet18 backbone.

---

> > ### Comment · Reviewer_EQbt · 2025-04-08
> >
> > Thanks for your feedback. It would be great for authors to include those updated results in the main paper.
> >
> > Following up on the point #1 raised: Thank you for the detailed explanation regarding the separate nature of the micro and macroscale analyses and the current limitations (data type/source mismatch). While a direct comparison isn't feasible now, could you speculate on the potential relationship between attribute effects at these different scales? For future work aiming to overcome the binary vs. continuous challenge, do you think employing generation methods based on descriptive inputs, like those using large language models, could be a viable path to create controlled binary variations at the micro-level, thus allowing for a more direct comparison with the binary macro-scale findings you presented?

---

> > > ### Author Response · Authors · 2025-04-09
> > >
> > > We thank the reviewer for the comment. If accepted, we will include the updated results in the main paper.
> > >
> > > We speculate that the attributes affecting the macroscale the most are less impactful on the microscale.
> > > With all the limitations already discussed and only referring to the analyzed models, we can describe the most discriminative attributes between identities as more impactful for the macroscale. At the same time, we can speculate that the model has learned to be invariant to the same attributes (e.g., gender) inside the embedding region corresponding to a single identity since they should remain approximately constant for a specific identity.
> > >
> > > We note that some attributes are continuous and can be finely discretized (face orientation), whereas others are inherently categorical (e.g., wearing eyeglasses).
> > > For continuous attributes, we further note that while observing the microscale requires relatively small variations of the attributes, the macroscale is generally affected by large variations since it looks at the embedding space across identities (e.g., gender change = "big" attribute variation). However, for some categorical attributes (wearing a necklace), we might speculate that the microscale is affected, whereas the macroscale is not. The invariance energy can capture this behavior inside the identity regions at the microscale.
> > > Note also that in the microscale experiments with GanControl, we applied a refined discretization of the latent space to compute the local displacements relative to the single attributes (see Appendix C.2), attaining identity retention on augmented data.
> > > During the design of our experiments, at a preliminary phase, we considered using LLM multimodal generative models for image generation to obtain controlled data augmentation. We found it challenging to obtain a programmatically controllable and reproducible augmentation providing a refined discretization representing local variations of intrinsically continuous attributes.
> > > On the other hand, at the macroscale, we think that using multimodal LLM to obtain augmented images is more manageable. However, it remains challenging to achieve controllable and reproducible data augmentation. We might suggest that to make the LLM generation more robust, it is generally possible to use a classifier to describe the attribute of the generated images.
> > >
> > > Nonetheless, we observe that the process of attribute definition, at least for face images, can be inherently multimodal for many face attributes because it can involve describing images through text. Thus, we agree that multimodal LLM for refined augmentation to analyze the embedding space at the microscale and the macroscale is a meaningful suggestion for future research to enable direct comparison at the two scales.
> > >
> > > As already pointed out to Rev 43PU, we think that the analysis of embeddings in FR can be a playground for the development of ML methods useful beyond the FR domain, including controllable augmentation via multimodal LLM.

---

### Decision · Program_Chairs · 2025-05-01

**Decision:**

Accept (poster)

**Comment:**

This paper was reviewed by 4 experts in the field who provided detailed suggestions. The reviewers’ recommendations were somewhat divergent, with EQbt recommending Accept, reviewers zixr, and MCoe recommending Weak Accept, and reviewer 43PU recommending Weak Reject.

The reviewers noted the paper's strengths in providing deeper interpretability of face recognition embeddings from a geometric perspective (MCoe) and proposing a novel metric (invariance measure/energy) to quantify model sensitivity to attributes, potentially guiding model improvement (EQbt, zixr, MCoe).

The reviewers did have some concerns, however, which included issues with clarity, presentation, and the interpretation of the results, particularly the proposed energy measure (43PU, zixr). The significance or novelty of some findings and the paper's relevance to ICML were also questioned (MCoe, 43PU). The authors provided a detailed rebuttal, which helped clarify some of details, causing 43PU to raise their initial recommendation from Reject to Weak Reject.

I read the reviews, rebuttal and feedback. I believe study of face embeddings is of value to the community, from the perspectives of both face rec. and interpretability, and accordingly side with the positive recommendation or Accept.